# A General Graph Spectral Wavelet Convolution via Chebyshev Order Decomposition

**Nian Liu** [1] **Xiaoxin He** [1] **Thomas Laurent** [2] **Francesco Di Giovanni** [3] **Michael M. Bronstein** [3 4] **Xavier Bresson** [1]

## Abstract

Spectral graph convolution, an important tool of data filtering on graphs, relies on two essential decisions: selecting spectral bases for signal transformation and parameterizing the kernel for frequency analysis. While recent techniques mainly focus on standard Fourier transform and vector-valued spectral functions, they fall short in flexibility to model signal distributions over large spatial ranges, and capacity of spectral function. In this paper, we present a novel wavelet-based graph convolution network, namely WaveGC, which integrates multi-resolution spectral bases and a matrix-valued filter kernel. Theoretically, we establish that WaveGC can effectively capture and decouple short-range and long-range information, providing superior filtering flexibility, surpassing existing graph wavelet neural networks. To instantiate WaveGC, we introduce a novel technique for learning general graph wavelets by separately combining odd and even terms of Chebyshev polynomials. This approach strictly satisfies wavelet admissibility criteria. Our numerical experiments showcase the consistent improvements in both short-range and long-range tasks. This underscores the effectiveness of the proposed model in handling different scenarios. Our code is available at `https://github.com/liun-online/WaveGC`.

## 1. Introduction

Spectral graph theory (SGT) (Chung, 1997), which enables analysis and learning on graph data, has firmly established itself as a pivotal methodology in graph machine learning. A significant milestone in SGT is the generalization of the convolution operation to graphs, as convolution for grid-structured data, i.e. sequences and images, has demonstrated remarkable success (LeCun et al., 1998; Hinton et al., 2012; Krizhevsky et al., 2012). Significant research interests in graph convolution revolve around two key factors: (1) *designing diverse bases for spectral transform*, and (2) *parameterizing powerful graph kernel*. For (1), the commonly used graph Fourier basis, consisting of the eigenvectors of the graph Laplacian (Shuman et al., 2013), stands as a prevalent choice. However, graph wavelets (Hammond et al., 2011) offer enhanced flexibility by constructing adaptable bases. For (2), classic approaches involve diagonalizing the kernel with fully free parameters (Bruna et al., 2013) or employing various polynomial approximations such as Chebyshev (Defferrard et al., 2016) and Cayley (Levie et al., 2018) polynomials. Additionally, convolution with a tensor-valued kernel serves as the spectral function of Transformer (Vaswani et al., 2017) under the shift-invariant condition (Li et al., 2021; Guibas et al., 2021).

Despite the existence of techniques in each aspect, the integration of these two lines into a unified framework remains challenging, impeding the full potential of graph convolution. In an effort to unravel this challenge, we introduce a novel operation — **Wave**let-based **G**raph **C**onvolution (WaveGC), which seamlessly incorporates both spectral basis and kernel considerations. In terms of spectral basis design, WaveGC is built upon graph wavelets, allowing it to capture information across the entire graph through a multi-resolution approach from highly adaptive construction of multiple graph wavelet bases. For filter parameterization, we opt for a matrix-valued spectral kernel with weight-sharing. The matrix-valued kernel offers greater flexibility to filter wavelet signals, thanks to its larger parameter space.

To comprehensively explore WaveGC, we theoretically analyse and assess its information-capturing capabilities. In contrast to the K-hop basic message-passing framework, WaveGC is demonstrated to exhibit both significantly larger and smaller receptive fields concurrently, achieved through the manipulation of scales. Previous graph wavelet theory (Hammond et al., 2011) only verifies the localization in small scale limit. Instead, our proof is complete as it covers both extremely small and large scales from the perspective

---

[1]National University of Singapore [2]Loyola Marymount University [3]University of Oxford [4]AITHYRA, Austria. Correspondence to: Nian Liu <nianliu@comp.nus.edu.sg>.

*Proceedings of the $42^{nd}$ International Conference on Machine Learning*, Vancouver, Canada. PMLR 267, 2025. Copyright 2025 by the author(s).

of information mixing (Di Giovanni et al., 2023). Moreover, our proof also implies that WaveGC is capable of simultaneously capturing both short-range and long-range information for each node, which facilitate global node interaction.

To implement WaveGC, a critical step lies in constructing graph wavelet bases that satisfy two fundamental criteria: (1) meeting the wavelet admissibility criteria (Mallat, 1999) and (2) showing adaptability to different graphs. Existing designs of graph wavelets face limitations, with some falling short in ensuring the criteria (Xu et al., 2019a; 2022), while others having fixed wavelet forms, lacking adaptability (Zheng et al., 2021; Cho et al., 2023). To address these limitations, we propose an innovative and general implementation of graph wavelets. Our solution involves *approximating scaling function basis and multiple wavelet bases using odd and even terms of Chebyshev polynomials, respectively*. This approach is inspired by our observation that, after a certain transformation, even terms of Chebyshev polynomials strictly satisfy the admissibility criteria, while odd terms supplement direct current signals. Through the combination of these terms via learnable coefficients, we aim to theoretically approximate scaling function and multiple wavelets with arbitrary complexity and flexibility. Our contributions are:

- We derive a new wavelet-based graph convolution (WaveGC), which integrates multi-resolution bases and matrix-valued kernel, enhancing spectral convolution on large spatial ranges.
- We theoretically prove that WaveGC can capture and distinguish the information from short and long ranges, surpassing conventional graph wavelet techniques.
- We pioneer an implementation of learnable graph wavelets, employing odd terms and even terms of Chebyshev polynomials individually. This implementation strictly satisfies the wavelet admissibility criteria.
- Our approach consistently outperforms baseline methods on both short-range and long-range tasks, achieving up to 15.7% improvement on VOC dataset.

## 2. Preliminaries

An undirected graph can be presented as $\mathcal{G} = (\mathcal{V}, E)$, where $\mathcal{V}$ is the set of $N$ nodes and $E \subseteq \mathcal{V} \times \mathcal{V}$ is the set of edges. The adjacency matrix of this graph is $\boldsymbol{A} \in \{0,1\}^{N \times N}$, where $\boldsymbol{A}_{ij} \in \{0,1\}$ denotes the relation between nodes $i$ and $j$ in $\mathcal{V}$. The degree matrix is $\boldsymbol{D} = \mathrm{diag}(d_1, \ldots, d_N) \in \mathbb{R}^{N \times N}$, where $d_i = \sum_{j \in \mathcal{V}} \boldsymbol{A}_{ij}$ is the degree of node $i \in \mathcal{V}$. The node feature matrix is $\boldsymbol{X} = [x_1, x_2, \ldots, x_N] \in \mathbb{R}^{N \times d_0}$, where $x_i$ is a $d_0$ dimensional feature vector of node $i \in \mathcal{V}$. Let $\hat{\boldsymbol{A}} = \boldsymbol{D}^{-\frac{1}{2}} \boldsymbol{A} \boldsymbol{D}^{-\frac{1}{2}}$ be the symmetric normalized adjacency matrix, then $\hat{\mathcal{L}} = \boldsymbol{I_N} - \hat{\boldsymbol{A}} = \boldsymbol{D}^{-\frac{1}{2}}(\boldsymbol{D} - \boldsymbol{A})\boldsymbol{D}^{-\frac{1}{2}}$ is the symmetric normalized

graph Laplacian. With eigen-decomposition, $\hat{\mathcal{L}} = \boldsymbol{U}\boldsymbol{\Lambda}\boldsymbol{U}^\top$, where $\boldsymbol{\Lambda} = \mathrm{diag}(\lambda_1, \ldots, \lambda_N) \in \mathbb{R}^{N \times N}, \lambda_i \in [0, 2]$ and $\boldsymbol{U} = [\boldsymbol{u_1}^\top, \ldots, \boldsymbol{u_N}^\top] \in \mathbb{R}^{N \times N}$ are the eigenvalues and eigenvectors of $\hat{\mathcal{L}}$, respectively. Given a signal $f \in \mathbb{R}^N$ on $\mathcal{G}$, the graph Fourier transform (Shuman et al., 2013) is defined as $\hat{f} = \boldsymbol{U}^\top f \in \mathbb{R}^N$, and its inverse is $f = \boldsymbol{U}\hat{f} \in \mathbb{R}^N$.

**Spectral graph wavelet transform (SGWT).** Hammond et al. (2011) redefine the wavelet basis (Mallat, 1999) on vertices in the spectral graph domain. Specifically, the SGWT is composed of three components: (1) *Unit wavelet basis*, denoted as $\Psi$ such that $\Psi = g(\hat{\mathcal{L}}) = \boldsymbol{U}g(\boldsymbol{\Lambda})\boldsymbol{U}^\top$, where $g$ acts as a band-pass filter $g : \mathbb{R}^+ \to \mathbb{R}^+$ meeting the following *wavelet admissibility criteria* (Mallat, 1999):

$$\mathcal{C}_\Psi = \int_{-\infty}^{\infty} \frac{|g(\lambda)|^2}{|\lambda|} d\lambda < \infty. \tag{1}$$

To meet this requirement, $g(\lambda = 0) = 0$ and $\lim_{\lambda \to \infty} g(\lambda) = 0$ are two essential prerequisites. (2) *Spatial scales*, a series of positive real values $\{s_j\}$ where distinct values of $s_j$ with $\Psi_{s_j} = \boldsymbol{U}g(s_j\boldsymbol{\Lambda})\boldsymbol{U}^\top$ can control different size of neighbors. (3) *Scaling function basis*, denoted as $\Phi$ such that $\Phi = \boldsymbol{U}h(\lambda)\boldsymbol{U}^\top$. Here, the function of $h : \mathbb{R}^+ \to \mathbb{R}^+$ is to supplement direct current (DC) signals at $\lambda = 0$, which is omitted by all wavelets $g(s_j\lambda)$ since $g(0) = 0$. Next, given a signal $f \in \mathbb{R}^N$, the formal SGWT (Hammond et al., 2011) is:

$$W_f(s_j) = \Psi_{s_j} f = \boldsymbol{U}g(s_j\boldsymbol{\Lambda})\boldsymbol{U}^\top f \in \mathbb{R}^N, \tag{2}$$

where $W_f(s_j)$ is the wavelet coefficients of $f$ under scale $s_j$. Similarly, scaling function coefficients are given by $S_f = \Phi f = \boldsymbol{U}h(\boldsymbol{\Lambda})\boldsymbol{U}^\top f \in \mathbb{R}^N$. Let $G(\lambda) = h(\lambda)^2 + \sum_j g(s_j\lambda)^2$, then if $G(\lambda) \equiv 1, \forall \lambda \in \boldsymbol{\Lambda}$, the constructed graph wavelets are known as *tight frames*, which guarantee energy conservation of the given signal between the original and the transformed domains (Shuman et al., 2015). More spectral graph wavelets are introduced in Appendix E.

## 3. From Graph Convolution to Graph Wavelets

Spectral graph convolution is a fundamental operation in the field of graph signal processing (Shuman et al., 2013). Specifically, given a signal matrix (or node features) $\boldsymbol{X} \in \mathbb{R}^{N \times d}$ on graph $\mathcal{G}$, the spectral filtering of this signal is defined with a kernel $\boldsymbol{\kappa} \in \mathbb{R}^{N \times N}$ by the convolution theorem (Arfken, 1985):

$$\boldsymbol{\kappa} *_{\mathcal{G}} \boldsymbol{X} = \mathcal{F}^{-1}(\mathcal{F}(\boldsymbol{\kappa}) \cdot \mathcal{F}(\boldsymbol{X})) \in \mathbb{R}^{N \times d}, \tag{3}$$

where $\cdot$ is the matrix multiplication operator, $\mathcal{F}(\cdot)$ and $\mathcal{F}^{-1}(\cdot)$ are the spectral transform (e.g., graph Fourier transform (Bruna et al., 2013)) and corresponding inverse transform, respectively. To implement a spectral convolution,

two critical choices must be considered in Eq. (3): 1) the selection of the transform $\mathcal{F}$ and 2) the parameterization of the kernel $\boldsymbol{\kappa}$.

## 3.1. General spectral wavelet via Chebyshev decomposition

For the selection of the spectral transform $\mathcal{F}$ and its inverse $\mathcal{F}^{-1}$, it can be tailored to the specific nature of data. For set data, the Dirac Delta function (Oppenheim et al., 1997) is employed, while the fast Fourier Transform (FFT) proves efficient for both sequences (Li et al., 2021) and grids (Guibas et al., 2021). In the context of graphs, the Fourier transform ($\mathcal{F} \to \boldsymbol{U}^{\top}$) emerges as one classical candidate. However, some inherent flaws limit the capacity of Fourier bases. (1) Standard graph Fourier bases, represented by one fixed matrix $\boldsymbol{U}^{\top}$, maintain a constant resolution and fixed frequency modes. (2) Fourier bases lack the adaptability to be further optimized according to different datasets and tasks. Therefore, *multiple resolution* and *adaptability* are two prerequisites for the design of an advanced base.

Notably, wavelet base is able to conform the above two demands, and hence offers enhanced filtering compared to Fourier base. For the resolution, the use of different scales $s_j$ allows wavelet to analyze detailed components of a signal at different granularities. More importantly, due to its strong spatial localization (Hammond et al., 2011), each wavelet corresponds to a signal diffused away from a central node (Xu et al., 2019a). Therefore, these scales also control varying receptive fields in spatial space, which enables the simultaneous fusion of short- and long-range information. For the adaptability, graph wavelets offer the flexibility to adjust the shapes of wavelets and scaling function. These components can be collaboratively optimized for the alignment of basis characteristics with different datasets, potentially enhancing generalization performance.

Next, we need to determine the form of the scaling function basis $\Phi = \boldsymbol{U}h(\boldsymbol{\Lambda})\boldsymbol{U}^{\top}$, the unit wavelet basis $\Psi = \boldsymbol{U}g(\boldsymbol{\Lambda})\boldsymbol{U}^{\top}$, and the scales $s_j$. The forms of $h$ and $g$ are expected to be powerful enough and easily available. Concurrently, $g$ should strictly satisfy the wavelet admissibility criteria, i.e., Eq. (1), and $h$ should complementarily provide DC signals. To achieve this target, we *separately introduce odd terms and even terms from Chebyshev polynomials (Hammond et al., 2011)* into the approximation of $h$ and $g$. Please recall that the Chebyshev polynomial $T_k(y)$ of order $k$ may be computed by the stable recurrence relation $T_k(y) = 2yT_{k-1}(y) - T_{k-2}(y)$ with $T_0 = 1$ and $T_1 = y$. After the following transform, we surprisingly observe that these transformed terms match all above expectations:

$$T_k(y) \to 1/2 \cdot (-T_k(y-1) + 1). \tag{4}$$

To give a more intuitive illustration, we present the spec-

tra of first six Chebyshev polynomials before and after the transform in Fig. 1 (b), where the set of odd and even terms after the transform are denoted as $\{T_i^o\}$ and $\{T_i^e\}$, respectively. From the figure, $g(\lambda = 0) \equiv 0$ for all $\{T_i^e\}$, and $h(\lambda = 0) \equiv 1$ for all $\{T_i^o\}$. Consequentially, $\{T_i^e\}$ and $\{T_i^o\}$ strictly meet the criteria and naturally serve as the basis of unit wavelet and scaling function. Moreover, not only can we easily get each Chebyshev term via iteration, but the constructed wavelet owns arbitrarily complex waveform because of the combination of as many terms as needed. Given $\{T_i^e\}$ and $\{T_i^o\}$, all we need to do is just to learn the coefficients to form the corresponding $g(\lambda)$ and $h(\lambda)$:

$$
\begin{aligned}
g(\boldsymbol{\Lambda}) &= \sum_i^{\rho} a_i T_i^e(\boldsymbol{\Lambda}) \in \mathbb{R}^{N \times N}, \\
h(\boldsymbol{\Lambda}) &= \sum_i^{\rho} b_i T_i^o(\boldsymbol{\Lambda}) \in \mathbb{R}^{N \times N},
\end{aligned}
\tag{5}
$$

where $\rho = K/2$ ($K$ is the total number of truncated Chebyshev terms), $\tilde{\boldsymbol{a}} = (a_1, a_2, \ldots, a_\rho) \in \mathbb{R}^{1 \times \rho}$ and $\tilde{\boldsymbol{b}} = (b_1, b_2, \ldots, b_\rho) \in \mathbb{R}^{1 \times \rho}$ represent two learnable coefficient vectors as follows:

$$\tilde{\boldsymbol{a}} = \text{Mean}(\boldsymbol{W_a}\hat{\boldsymbol{Z}} + \boldsymbol{b_a}), \quad \tilde{\boldsymbol{b}} = \text{Mean}(\boldsymbol{W_b}\hat{\boldsymbol{Z}} + \boldsymbol{b_b}), \tag{6}$$

where $\{\boldsymbol{W_a}, \boldsymbol{W_b}\} \in \mathbb{R}^{d \times \rho}$ and $\{\boldsymbol{b_a}, \boldsymbol{b_b}\} \in \mathbb{R}^{1 \times \rho}$ are learnable parameters, and $\hat{\boldsymbol{Z}}$ is the eigenvalue embedding composed by the module in (Bo et al., 2023). Further details can be found in Appendix B. Also, we can learn the scales $\tilde{\boldsymbol{s}} = (s_1, s_2, \ldots, s_J)$ in the same way:

$$\tilde{\boldsymbol{s}} = \sigma(\text{Mean}(\boldsymbol{W_s}\hat{\boldsymbol{Z}} + \boldsymbol{b_s})) \cdot \overline{\boldsymbol{s}} \in \mathbb{R}^{1 \times J}, \tag{7}$$

where $\sigma$ is sigmoid function, $\boldsymbol{W_s} \in \mathbb{R}^{d \times J}$ and $\boldsymbol{b_s} \in \mathbb{R}^{1 \times J}$ are learnable parameters, and $\overline{\boldsymbol{s}} = (\overline{s_1}, \overline{s_2}, \ldots, \overline{s_J})$ is a predefined vector to control the size of $\tilde{\boldsymbol{s}}$.

Based on our construction, $g(\lambda)$ is a strict band-pass filter in [0, 2], while $s$ can scale its shape in $g(s\lambda)$. Specifically, $s < 1$ "stretches" the shape of $g(\lambda)$, and $s > 1$ "squeezes" its shape (Please refer to Fig. 9). To maintain the same spectral interval [0, 2], we truncate $g(s\lambda)$ within the intersection of $\lambda \in [0, 2]$ and $\lambda \in [0, 2/s]$.

## 3.2. Matrix-valued kernel via weight sharing

Next, we consider the parametrization of the convolutional kernel $\mathcal{F}(\boldsymbol{\kappa})$. In the spectral domain, each Fourier mode typically corresponds to a global frequency pattern, either low- or high-frequency. Consequently, in Fourier-based approaches, it is common to apply a vector-valued kernel over the diagonalized graph Laplacian spectrum, denoted as diag($\theta_\lambda$) (Bruna et al., 2013; Defferrard et al., 2016; Levie et al., 2018), which effectively scales these global frequency components. However, this strategy becomes unsuitable after applying a wavelet transform. Unlike Fourier

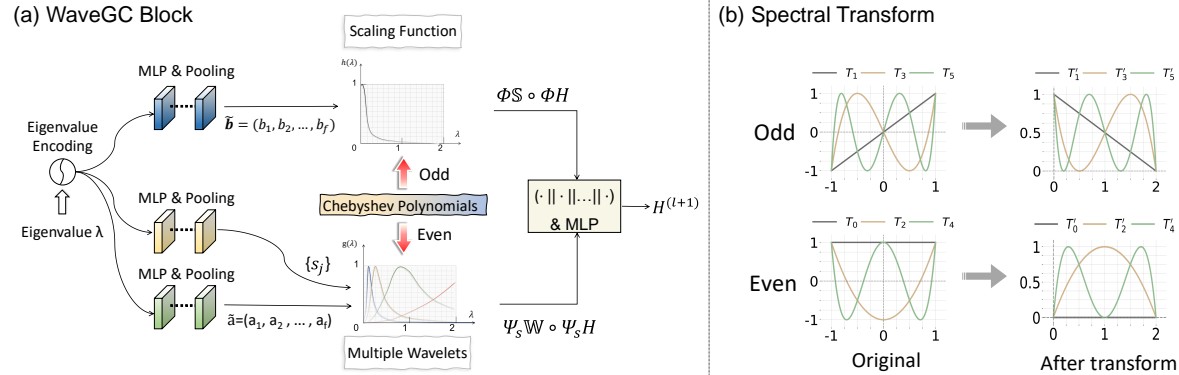

*Figure 1.* (a) Overview of our proposed WaveGC technique. (b) Illustration of Chebyshev polynomials before and after the given transform, from [-1, 1] to [0, 2]. In this representation, we distinguish odd and even terms, presenting only the first three terms for each.

bases, wavelet coefficients encode localized, node-specific patterns that may capture short- or long-range interactions, but not global frequency modes. As a result, a different parametrization scheme, tailored to the localized nature of wavelet representations, is required.

Along another line of research, Fourier Neural Operator (FNO) (Li et al., 2021) models the convolution kernel as a fully learnable tensor $\mathbb{M} \in \mathbb{R}^{N \times d \times d}$, where $N$ is the number of frequency modes, and $d$ is the feature dimension. This tensor-valued kernel offers two notable advantages. First, although FNO was originally introduced in the context of the Fourier transform, the kernel $\mathbb{M}$ is inherently independent of graph spectrum, and is thus amenable to generalization across other transforms (Tripura & Chakraborty, 2023). Second, in contrast to vector-valued kernels, the matrix-valued formulation provides a significantly larger number of learnable parameters, thereby increasing its expressivity and capacity to adapt to complex patterns. Experimental results presented in Section 6.2 empirically demonstrate that the matrix-valued kernel outperforms its vector-valued counterpart in the context of filtering wavelet-transformed signals.

In this paper, we adopt the tensor $\mathbb{M}$ for the convolution kernel. The standard parameter count for $\mathbb{M}$ is $N \times d \times d$. This can lead to a substantial number of parameters, especially for large-scale graphs with high $N$, increasing the risk of overfitting. To mitigate this while preserving model expressivity, we introduce a parameter-sharing strategy across all frequency modes by employing a single MLP. This approach reduces the number of learnable parameters from $N \times d \times d$ (tensor) to $d \times d$ (matrix). Accordingly, the convolution operation in Eq. (3) simplifies to $\mathbb{M} *_{\mathcal{G}} \boldsymbol{X} = \mathcal{F}^{-1} \mathbb{M} \circ \mathcal{F}(\boldsymbol{X}) = \mathcal{F}^{-1}(\text{MLP}(\mathcal{F}(\boldsymbol{X})))$, where $\circ$ is the composition between two functions. An alternative method is presented in AFNO (Guibas et al., 2021), introducing a similar technique that offers improved efficiency but with a more intricate design.

### 3.3. WAVELET-BASED GRAPH CONVOLUTION

Until now, we have elaborated the proposed advancements on kernel and bases, and now discuss how to integrate these two aspects. Provided that we have $J$ wavelet $\{\Psi_{s_j}\}_{j=1}^J$ and one scaling function $\Phi$ constructed via the above Chebyshev decomposition, $\mathcal{F}: \mathbb{R}^{N \times d} \to \mathbb{R}^{N(J+1) \times d}$ in Eq. (3) is the stack of transforms from each component:

$$\mathcal{F}(\boldsymbol{H}^{(l)}) = \boldsymbol{T}\boldsymbol{H}^{(l)} = ((\Phi \boldsymbol{H}^{(l)})^{\top} \|$$
$$(\Psi_{s_1} \boldsymbol{H}^{(l)})^{\top} \| ... \| (\Psi_{s_J} \boldsymbol{H}^{(l)})^{\top})^{\top} \in \mathbb{R}^{N(J+1) \times d}, \tag{8}$$

where $\boldsymbol{T} = (\Phi^{\top} \| \Psi_{s_1}^{\top} \| ... \| \Psi_{s_J}^{\top})^{\top}$ is the overall transform and $\|$ means concatenation, $\boldsymbol{H}^{(l)}$ is the node embedding matrix at layer $l$. Next, we check if the inverse $\mathcal{F}^{-1}$ exists. Considering $\boldsymbol{T}$ is not a square matrix, $\mathcal{F}^{-1}$ should be its pseudo-inverse as $(\boldsymbol{T}^{\top}\boldsymbol{T})^{-1}\boldsymbol{T}^{\top}$, where $\boldsymbol{T}^{\top}\boldsymbol{T} = \Phi\Phi^{\top} + \sum_{j=1}^J \Psi_{s_j}\Psi_{s_j}^{\top} = \boldsymbol{U}[h(\lambda)^2 + \sum_{j=1}^J g(s_j\lambda)^2]\boldsymbol{U}^{\top}$. Ideally, if $\boldsymbol{T}$ is imposed as **tight frames**, then $h(\lambda)^2 + \sum_{j=1}^J g(s_j\lambda)^2 = \boldsymbol{I}$ (Leonardi & Van De Ville, 2013), and $\boldsymbol{T}^{\top}\boldsymbol{T} = \boldsymbol{U}\boldsymbol{I}\boldsymbol{U}^{\top} = \boldsymbol{I}$. In this case, $\mathcal{F}^{-1} = (\boldsymbol{T}^{\top}\boldsymbol{T})^{-1}\boldsymbol{T}^{\top} = \boldsymbol{T}^{\top}$, and Eq. (3) becomes:

$$\boldsymbol{H}^{(l+1)} = \boldsymbol{T}^{\top}\mathbb{M} \circ \boldsymbol{T}\boldsymbol{H}^{(l)}$$
$$= \Phi\mathbb{S} \circ \Phi\boldsymbol{H}^{(l)} + \sum_{j=1}^J \Psi_{s_j}\mathbb{W}_j \circ \Psi_{s_j}\boldsymbol{H}^{(l)} \in \mathbb{R}^{N \times d}, \tag{9}$$

where we separate $\mathbb{M}$ into $\mathbb{S}$ and $\{\mathbb{W}_j\}_{j=0}^J$ as scaling kernel and different wavelet kernels.

**How to guarantee tight frames?** From above derivations, *tight frames* is a key for the simplification of inverse $\mathcal{F}^{-1}$ in Eq. (9). This can be guaranteed by $l_2$ norm on the above constructed wavelets and scaling function. For each eigenvalue $\lambda_i \in \Lambda$, we have $v_i^2 = h(\lambda_i)^2 + \sum_{j=1}^J g(s_j\lambda_i)^2$, $\tilde{h}(\lambda_i) = h(\lambda_i)/v$, $\tilde{g}_i(s_j\lambda_i) = g(s_j\lambda_i)/v$. Then, $G(\Lambda) = \tilde{h}(\Lambda)^2 + \sum_j \tilde{g}(s_j\Lambda)^2 = \boldsymbol{I}$ forms tight frames (Section 2). Thus, while the pseudo-inverse must theoretically exist, we

Table 1. Comparison between spectral graph convolution and WaveGC.

|  | Spectral Graph Convolution | WaveGC |
|---|---|---|
| Kernel | $\text{diag}(\theta_\lambda)$: Diagonal matrix | $\mathbb{S} \,/\, \mathbb{W}$: Full matrix |
| Bases | $\mathbf{U}^\top$: Fourier basis | $\Phi \,/\, \Psi_s$: Scaling / Wavelet basis |
| Convolution | $\mathbf{U}\text{diag}(\theta_\lambda)\mathbf{U}^\top\mathbf{X}$ | $\Phi\mathbb{S} \circ \Phi\mathbf{X} \,/\, \Psi_s\mathbb{W} \circ \Psi_s\mathbf{X}$ |

can circumvent the necessity of explicitly calculating the pseudo-inverse.

Resembling the multi-head attention (Vaswani et al., 2017), we treat each wavelet transform as a "wavelet head", and concatenate them rather than sum them to get $\boldsymbol{H}^{(l+1)} \in \mathbb{R}^{N \times d}$:

$$
\begin{aligned}
\boldsymbol{H}^{(l+1)} = \sigma \Big( \Big[ \Phi\mathbb{S} \circ \Phi\boldsymbol{H}^{(l)} || \Psi_{s_1}\mathbb{W}_1 \circ \Psi_{s_1}\boldsymbol{H}^{(l)} || \\
\dots || \Psi_{s_J}\mathbb{W}_J \circ \Psi_{s_J}\boldsymbol{H}^{(l)} \Big] \cdot \boldsymbol{W} \Big),
\end{aligned}
\tag{10}
$$

where an outermost MLP increases the flexibility. Fig. 1 (a) presents the whole framework of our wavelet-based graph convolution, or WaveGC. For a better understanding, we compare spectral graph convolution and WaveGC in Table. 1, where WaveGC contains only one wavelet for simplicity. Based on the differences shown in the table, WaveGC endows spectral graph convolution with the beneficial inductive bias of long-range dependency.

## 4. Theoretical Properties of WaveGC

Traditionally, wavelet is notable for its diverse receptive fields because of varying scales (Mallat, 1999). For graph wavelet, Hammond et al. (2011) were the first to prove the localization when scale $s \to 0$, but did not discuss the long-range case when $s \to \infty$. We further augment this discussion and demonstrate the effectiveness of the proposed WaveGC in capturing both short- and long-range information. Intuitively, a model's ability to integrate global information enables the reception and mixing of messages from distant nodes. Conversely, a model with a limited receptive field can only effectively mix local messages. Hence, assessing the degree of information 'mixing' becomes a key property. For this reason, we focus on the concept of *maximal mixing*:

**Definition 4.1. (Maximal mixing)** (Di Giovanni et al., 2023). *For a twice differentiable graph-function $y_G$ of node features $\boldsymbol{x}_i$, the maximal mixing induced by $y_G$ among the features $\boldsymbol{x}_a$ and $\boldsymbol{x}_b$ with nodes $a, b$ is*

$$
\text{mix}_{y_G}(a, b) = \max_{\boldsymbol{x}_i} \max_{1 \le \alpha, \beta \le d} \left| \frac{\partial^2 y_G(\boldsymbol{X})}{\partial x_a^\alpha \partial x_b^\beta} \right|. \tag{11}
$$

This definition is established in the context of graph-level task, and $y_G$ is the final output of an end-to-end framework,

comprising the primary model and a readout function (e.g., mean, max) applied over the last layer. $\alpha$ and $\beta$ represent two entries of the $d$-dimensional features $\boldsymbol{x}_a$ and $\boldsymbol{x}_b$.

Next, we employ the concept of 'maximal mixing' on the WaveGC. For simplicity, we only take one wavelet basis $\Psi_s$ for analysis. The capacity of $\Psi_s$ on mixing information depends on two factors, i.e. $K$-order Chebyshev term and scale $s$. For a fair discussion on the effect of $s$ on message passing, we compare $\sigma(\Psi_s HW)$ and K-order message passing with the form of $\sigma(\sum_{j=0}^{K} \tau_j A^j HW)$, $\tau_j \in [0, 1]$:

**Theorem 4.2 (Short-range and long-range receptive fields).** *Given a large even number $K > 0$ and two random nodes $a$ and $b$, if the depths $m_\Psi$ and $m_A$ are necessary for $\sigma(\Psi_s HW)$ and $\sigma(\sum_{j=0}^{K} \tau_j A^j HW)$ to induce the same amount of mixing $\text{mix}_{y_G}(b, a)$, then the lower bounds of $m_\Psi$ and $m_A$, i.e. $L_{m_\Psi}$ and $L_{m_A}$, approximately satisfy the following relation when scale $s \to 0$:*

$$
L_{m_\Psi} \approx \frac{P}{K} L_{m_A} + \frac{2|E|}{K\sqrt{d_a d_b}} \frac{\text{mix}_{y_G}(b, a)}{\gamma} \cdot \frac{1}{(\alpha^2 s^{2K})^{m_\Psi}}. \tag{12}
$$

*Or, if $s \to \infty$, the relation becomes:*

$$
L_{m_\Psi} \approx \frac{P}{K} L_{m_A} - \frac{2|E|}{K(K+1)^{2m_A}\tau_P^{2m_A}\sqrt{d_a d_b}} \frac{\text{mix}_{y_G}(b, a)}{\gamma}, \tag{13}
$$

*where $P < K$ and $(\tau_P A^P)_{ba} = \max\{(\tau_m A^m)_{ba}\}_{m=0}^{K}$. $d_a$ and $d_b$ are degrees of two nodes, and $\alpha = \frac{C \cdot 2^K (K+1)}{K!}$. $\gamma = \sqrt{\frac{d_{max}}{d_{min}}}$, where $d_{max}/d_{min}$ is the maximum / minimum degree in the graph.*

The proof is provided in Appendix A.3. In Eq. (12), since the second term on the right-hand side is large ($s \to 0$), it required $\Psi_s$ to propagate more layers to mix the nodes. Conversely, if $s \to \infty$ (Eq. (13)), $\Psi_s$ will achieve the same degree of node mixing as $K$-hop message passing but with less propagation. Moreover, the greater the "mixing" $\text{mix}_{y_G}(b, a)$ is required between nodes, the fewer number of layers $L_{m_\Psi}$ is needed compared to $L_{m_A}$. To conclude, $\Psi_s$ presents the short- and long-range characteristics of WaveGC on message passing, while these characteristics do not derive from the order $K$ of Chebyshev polynomials but from the scale $s$ exclusively.

*Table 2.* Qualified results on short-range tasks compared to baselines. **Bold**: Best, Underline: Runner-up, OOM: Out-of-memory. All results are reproduced based on source codes.

| Model | CS | Photo | Computer | CoraFull | ogbn-arxiv |
|---|---|---|---|---|---|
| | Accuracy ↑ | Accuracy ↑ | Accuracy ↑ | Accuracy ↑ | Accuracy ↑ |
| GCN | 92.92±0.12 | 92.70±0.20 | 89.65±0.52 | 61.76±0.14 | 71.74±0.29 |
| GAT | 93.61±0.14 | 93.87±0.11 | 90.78±0.13 | 64.47±0.18 | 71.82±0.23 |
| APPNP | 94.49±0.07 | 94.32±0.14 | 90.18±0.17 | 65.16±0.28 | 71.90±0.25 |
| Scattering | 94.77±0.33 | 92.10±0.61 | 85.68±0.71 | 57.65±0.84 | 66.23±0.19 |
| Scattering GCN | 95.18±0.30 | 93.07±0.42 | 88.83±0.44 | 61.14±1.13 | 71.18±0.76 |
| SGWT | 94.81±0.23 | 92.45±0.62 | 85.19±0.59 | 55.04±1.12 | 69.08±0.30 |
| GWNN | 90.75±0.59 | 94.45±0.45 | 90.75±0.59 | 64.19±0.79 | 71.13±0.47 |
| UFGConvS | 95.33±0.27 | 93.98±0.59 | 88.68±0.39 | 61.25±0.93 | 70.04±0.22 |
| UFGConvR | 95.46±0.33 | 94.34±0.34 | 89.29±0.46 | 62.43±0.80 | 71.97±0.12 |
| WaveShrink-ChebNet | 94.90±0.30 | 93.54±0.90 | 88.20±0.65 | 58.98±0.69 | OOM |
| DEFT | 95.04±0.32 | 94.35±0.44 | 91.63±0.52 | 68.01±0.86 | 72.01±0.20 |
| WaveNet | 94.91±0.29 | 94.09±0.63 | 92.06±0.33 | 57.65±1.05 | 71.37±0.14 |
| SEA-GWNN | 95.11±0.37 | 94.35±0.50 | 89.88±0.64 | 66.74±0.79 | 72.64±0.21 |
| **WaveGC** (ours) | **95.89±0.34** | **95.37±0.44** | **92.26±0.18** | **69.14±0.78** | **73.01±0.18** |

## 5. Why do we need decomposition?

As shown in Fig. 1 (b), odd and even terms of Chebyshev polynomials meet the requirements on constructing wavelet after decomposition and transform. Additionally, each term is apt to be obtained according to the iteration formula, while infinite number of terms guarantee the expressiveness of the final composed wavelet. Next, we compare our decomposition solution with other related techniques:

● *Constructing wavelet via Chebyshev polynomials.* Previous wavelet-based GNNs leverage Chebyshev polynomials with two purposes. (1) Approximate wavelets of pre-defined forms. SGWT (Hammond et al., 2011), GWNN (Xu et al., 2019a) and UFGConvS/R (Zheng et al., 2021) follow this line. They firstly fix the shape of wavelets as cubic spline, exponential or high-pass/low-pass filters, followed by the approximation via Chebyshev polynomials. In this pipeline, wavelet fails to learn further and suit the dataset and task at hand. (2) Compose a new wavelet. DEFT (Bastos et al., 2023) employs an MLP or GNN network to freely learn the coefficients before each Chebyshev basis. These coefficients are optimized according to the training loss, but loose the constraint on wavelet admissibility criteria.

● *No decomposition.* If we uniformly learn the coefficients for all Chebyshev terms without decomposition, WaveGC degrades to a variant similar to ChebNet (Defferrard et al., 2016). However, mixture rather than decomposition blends the signals from different ranges, and the final spatial ranges cannot be precisely predicted and controlled.

We provide numerical comparison and spectral visualization in section 6.3 for WaveGC against these related studies.

## 6. Numerical Experiments

In this section, we evaluate the performance of WaveGC on both short-range and long-range benchmarks using the

following datasets: (1) *Datasets for short-range tasks:* `CS`, `Photo`, `Computer` and `CoraFull` from the PyTorch Geometric (PyG) (Fey & Lenssen, 2019), and one large-size graph, i.e. `ogbn-arxiv` from Open Graph Benchmark (OGB) (Hu et al., 2020) (2) *Datasets for long-range tasks:* `PascalVOC-SP (VOC)`, `PCQM-Contact (PCQM)`, `COCO-SP (COCO)`, `Peptides-func (Pf)` and `Peptides-struct (Ps)` from LRGB (Dwivedi et al., 2022). Please refer to Appendix C.1 for implementation details and Appendix C.2 for details of datasets.

### 6.1. Benchmarking WaveGC

For short-range (S) datasets, we follow the settings from (Chen et al., 2022). For `ogbn-arxiv`, we use the public splits in OGB (Hu et al., 2020). For long-range datasets, we adhere to the experimental configurations outlined in (Dwivedi et al., 2022). The selected baselines belong to four categories, i.e., classical GNNs {GCN (Kipf & Welling, 2017), GAT (Velickovic et al., 2017), APPNP (Gasteiger et al., 2018), GINE (Xu et al., 2019b) and GatedGCN (Bresson & Laurent, 2017)}, graph scattering network {Scattering (Gama et al., 2018) and Scattering GCN (Min et al., 2020)}, spectral graph wavelet network {SGWT (Hammond et al., 2011), GWNN (Xu et al., 2019a), UFGConvS (Zheng et al., 2021), UFG-ConvR (Zheng et al., 2021), WaveShrink (Wan et al., 2023), DEFT (Bastos et al., 2023) and WaveNet (Yang et al., 2024)} and wavelet lifting transform {SEA-GWNN (Deb et al., 2024)} [1]. The results of the comparison with SOTA models are shown in Table 2 and 3, where our WaveGC demonstrates the best results on all datasets. Remarkably, the improvement on VOC achieves up to 11.83%, implying the superior long-range information perception.

In the experiments conducted on the five short-range

---

[1] For SGWT and Scattering, we concatenate the filtered signals from all bases and apply an MLP to get the final embeddings.

*Table 3.* Qualified results on long-range tasks compared to baselines. **Bold**: Best, Underline: Runner-up, OOM: Out-of-memory, All results are reproduced based on source codes.

| Model | VOC | PCQM | COCO | Pf | Ps |
|---|---|---|---|---|---|
| | F1 score ↑ | MRR ↑ | F1 score ↑ | AP ↑ | MAE ↓ |
| GCN | 12.68±0.60 | 32.34±0.06 | 08.41±0.10 | 59.30±0.23 | 34.96±0.13 |
| GINE | 12.65±0.76 | 31.80±0.27 | 13.39±0.44 | 54.98±0.79 | 35.47±0.45 |
| GatedGCN | 28.73±2.19 | 32.18±0.11 | 26.41±0.45 | 58.64±0.77 | 34.20±0.13 |
| Scattering | 16.58±0.49 | 33.90±0.27 | 16.44±0.79 | 56.80±0.38 | 26.77±0.11 |
| Scattering GCN | 30.45±0.36 | 33.73±0.45 | 30.27±0.60 | 62.87±0.64 | 26.43±0.20 |
| SGWT | 31.22±0.56 | 34.04±0.05 | 32.97±0.53 | 60.23±0.27 | 25.39±0.21 |
| GWNN | 25.60±0.56 | 32.72±0.08 | 13.39±0.44 | 65.47±0.48 | 27.34±0.04 |
| UFGConvS | 31.27±0.39 | 33.94±0.24 | 23.15±0.55 | 65.83±0.75 | 27.08±0.58 |
| UFGConvR | 31.08±0.33 | 34.08±0.20 | 26.02±0.48 | 65.29±0.82 | 27.50±0.21 |
| WaveShrink-ChebNet | 18.80±0.85 | 32.56±0.11 | 11.12±0.46 | 61.12±0.53 | 27.45±0.06 |
| DEFT | 35.98±0.20 | 34.25±0.06 | 30.14±0.49 | 66.95±0.63 | 25.06±0.13 |
| WaveNet | 28.60±0.15 | 33.19±0.20 | 23.06±0.18 | 64.63±0.27 | 25.88±0.01 |
| SEA-GWNN | 31.97±0.55 | 29.89±0.26 | 24.33±0.23 | 68.75±0.20 | 25.64±0.31 |
| **WaveGC** (ours) | **41.63±0.19** | **34.50±0.02** | **35.96±0.22** | **69.73±0.43** | **24.83±0.11** |

datasets, the model is required to prioritize local information, while the five long-range datasets necessitate the handling of distant interactions. The results clearly demonstrate that the proposed WaveGC consistently outperforms traditional graph convolutions and graph wavelets in effectively aggregating both local and long-range information.

### 6.2. Effectiveness of matrix-valued kernel

The proposed matrix-valued kernel and weight-sharing strategy mark an advancement over conventional graph convolution, particularly in the context of processing wavelet-based signals. In this section, we conduct a comprehensive analysis of the effectiveness of these two architectural innovations.

As shown in Table 4, the matrix-valued kernel consistently outperforms its vector-valued counterpart. This improvement suggests that increasing the expressiveness of the kernel—through a higher parameter capacity—enhances the model's ability on feature learning.

*Table 4.* Compare *Matrix-valued* and *Vector-valued* kernels.

| Kernel | Computer (Accuracy ↑) | Ps (MAE ↓) |
|---|---|---|
| *Vector-valued* | 89.96 | 25.30 |
| *Matrix-valued* | **92.26** | **24.83** |

In addition, Table 5 examines the impact of weight sharing across spectral frequencies. Assigning distinct kernels to individual frequencies does not improve performance and, even results in degradation. This decline is likely due to overfitting caused by the large number of parameters introduced in the non-sharing setup. Specifically, non-sharing kernels require a mapping from each eigenvalue embedding to a unique transformation matrix, defined as $f : \mathbb{R}^d \to \mathbb{R}^{d \times d}$, which is implemented using a multi-layer perceptron (MLP) with a weight dimension of $\mathbb{R}^{d \times d \times d}$, $d$ is the embedding dimension. For instance, when d=96 in `Ps`, this results in

an approximate increase about 876K parameters.

*Table 5.* Compare *sharing* and *non-sharing* kernel weights.

| Result (Parameters) | Computer (Accuracy ↑) | Ps (MAE ↓) |
|---|---|---|
| *Non-sharing* | 90.51 (535k) | 26.22 (1,410k)) |
| ***Sharing*** | **92.26 (167k)** | **24.83 (534k)** |

### 6.3. Effectiveness of learnable wavelet bases

In this section, we compare the learnt wavelet bases from WaveGC with other baselines, including five graph wavelets (i.e. SGWT (Hammond et al., 2011), UFGConvS/R (Zheng et al., 2021), DEFT (Bastos et al., 2023), GWNN (Xu et al., 2019a) and WaveNet (Yang et al., 2024)). We additionally evaluate ChebNet*, a variant of our WaveGC where the only change is to combine odd and even terms without decomposition. Therefore, the improvement of WaveGC over ChebNet* reflects the effectiveness of decoupling operation. The numerical comparison on `Computer` and `PascalVOC-SP` has been shown in Table. 2 and 3, which demonstrates obvious gains from WaveGC especially on long-range `PascalVOC-SP`. The ChebNet* gets 89.85 and 36.45 separately on `Computer` and `PascalVOC-SP`, still inferior to WaveGC. [2]

To address the performance gap observed on the VOC dataset, we provide insights through the spectral visualization of various bases in Fig. 2. Upon examination of various wavelets, those from SGWT and UFGConvS/R meet admissibility criteria with multiple resolutions, but these lines are not adaptive. DEFT outputs multiple bases with unpredictable shapes, so it is hard to strictly restrain these outputs as wavelets. GWNN adopts one exponential wavelet base, omitting information from different ranges as well as not meeting criteria. WaveNet and ChebNet* blend local and distant information in spatial space, hampering the deci-

---

[2]We explore more differences between ChebNet (Defferrard et al., 2016) and our WaveGC in Appendix C.4.

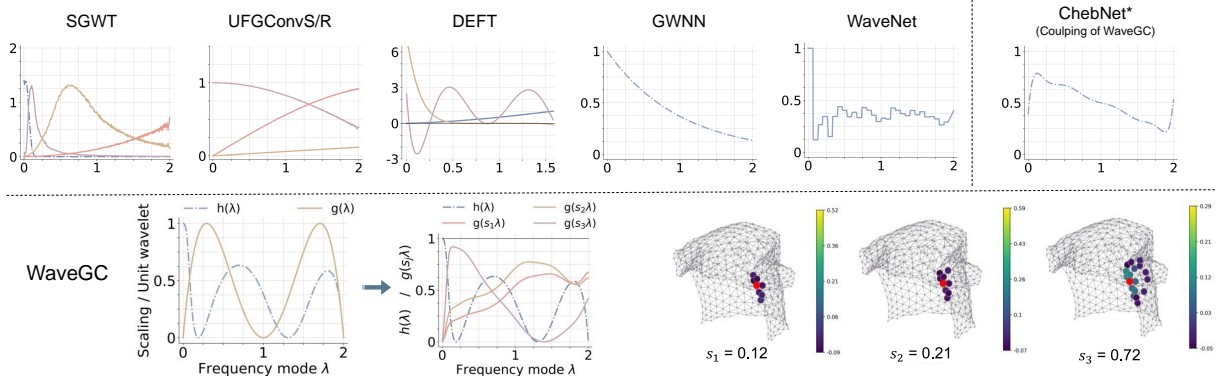

*Figure 2.* The spectral and spatial visualization of different bases on `PascalVOC-SP`.

sion on the best range. For our WaveGC, Fig. 2 intuitively demonstrates that the unit wavelet got by our decoupling of Chebyshev polynomials strictly meets the admissibility criteria, as Eq. (1), while the corresponding base scaling function supplements the direct current signals at $\lambda = 0$. After integration of learnable scales, the final wavelets also meet criteria and adapt to the demand on multiresolution. The plot of $G(\lambda) = h(\lambda)^2 + \sum_{j=1}^{3} g(s_j\lambda)^2$ as a black dashed line (located at 1) confirms the construction of tight frames via normalization technique. Fig.2 also depicts the signal distribution over the topology centered on the target node (the red-filled circle). As the scale $s_j$ increases, the receptive field of the central node expands. Once again, this visualization intuitively confirms the capability of WaveGC to aggregate both short- and long-range information simultaneously but distinguishingly. More analyses are given in Appendix C.3.1.

### 6.4. Ablation study

*Table 6.* Results of the ablation study. **Bold**: Best.

| Variants | Computer | Ps |
|---|---|---|
| | Accuracy ↑ | MAE ↓ |
| WaveGC | **92.26** | **24.83** |
| w/o wavelet | 89.65 | 34.20 |
| w/o MPNN | 90.89 | 25.04 |
| w/o $h(\lambda)$ | 90.57 | 25.12 |
| w/o $g(s\lambda)$ | 90.87 | 25.09 |

In this section, we conduct an ablation study of our WaveGC to assess the effectiveness of each component, and the corresponding results are presented in Table 6. The evaluation is conducted on `Peptides-struct` (long-range) and `Computer` (short-range).

Given the hybrid network (Fig. 3), we firstly remove the MPNN part (i.e., 'w/o MPNN') and wavelet part (i.e., 'w/o wavelet'), respectively. Both ablations degrade model performances, where 'w/o wavelet' decline more. To avoid interference from MPNN part, we base on 'w/o MPNN', and

continue to exclude scaling term (i.e., 'w/o $h(\lambda)$'), wavelet terms (i.e., 'w/o $g(s\lambda)$') and tight frame constrains (i.e., 'w/o tight frame'). Then, both the scaling function basis $h(\lambda)$ and wavelet bases $g(s\lambda)$ are essential components of our WaveGC. In particular, neglecting $h(\lambda)$ results in a larger drop in performance on both short-range and long-range cases, emphasizing the crucial role of low-frequency information.

### 6.5. Complexity analysis

The main complexity of WaveGC is the eigen-decomposition process, involving $O(N^3)$. This is practical for the small-to-medium graphs used in all long-range and some short-range benchmarks, where detailed spectral modeling is critical. To accelerate the decomposition on large-scale graph (e.g., ogbn-arxiv), we may adopt randomized SVD (Halko et al., 2009) with complexity $O(N^2 \log K)$, where we only pick the top $K$ eigenvectors.

*Table 7.* Training and EVD time on short- and long-range datasets.

| Short-range | CS | Photo | Computer | CoraFull | ogbn-arxiv |
|---|---|---|---|---|---|
| Training (min) | 5.70 | 0.95 | 4.87 | 22.00 | 36.67 |
| EVD (min) | 2.82 | 0.32 | 1.44 | 3.49 | 21.69 |
| Long-range | VOC | PCQM | COCO | Pf | Ps |
| Training (h) | 4.02 | 12.33 | 45.40 | 1.88 | 1.32 |
| EVD (h) | 0.05 | 0.21 | 0.58 | 0.02 | 0.02 |

Fig. 7 presents a direct comparison of the training time and EVD time across both short-range and long-range datasets. As shown, the time required for EVD is consistently lower than that of training across all datasets, with the difference being particularly significant in the long-range cases. Furthermore, the EVD operation is performed only once before training, and it is a prerequisite for most graph wavelet baselines. To further reduce complexity, we propose a fully polynomial-based approximation that removes the need for EVD, achieving total complexity of $O(N)$. More details are given in Appendix D.

**Other experiments** In Appendix C.5, we analyze the complexity and report the running time for WaveGC and other spectral graph wavelets. Our model shows shorter running times than competitive spectral models while being significantly more accurate. In Appendix C.6, we test the sensitivity of two important hyper-parameters.

## 7. Conclusion

In this study, we proposed a novel graph convolution operation based on wavelets (WaveGC), establishing its theoretical capability to capture information at both short and long ranges through a multi-resolution approach.

## Acknowledgements

XB is supported by NUS Grant R-252-000-B97-133 and MOE AcRF T1 Grant 251RES2423. MB is partially supported by the EPSRC Turing AI World-Leading Research Fellowship No. EP/X040062/1 and EPSRC AI Hub No. EP/Y028872/1. The authors would like to express their gratitude to the reviewers for their feedback, which has improved the clarity and contribution of the paper.

## Impact Statement

This paper presents work whose goal is to advance the field of Machine Learning. There are many potential societal consequences of our work, none which we feel must be specifically highlighted here.

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

# A. Theoretical Proof

Firstly, we give two auxiliary but indispensable lemma and theorem. Let starts from the formula $\sigma(\Psi_s HW)$. In this equation, we bound the first derivate of non-linear function as $|\sigma'| < c_\sigma$, and set $||W|| \leq w$, where $|| \cdot ||$ is the operator norm. First, we give an upper bound for each entry in $\Psi_s$.

**Lemma A.1** (**Upper bound for graph wavelet**). *Let* $\Psi = Ug(\Lambda)U^T$. *Given a large even number* $K > 0$, *then for* $\forall i, j \in V \times V$, *we have:*

$$(\Psi_s)_{ij} < \left(\alpha(\hat{A})^{K/2}s^K\right)_{ij}, \quad \alpha = \frac{C \cdot 2^K(K+1)}{K!}. \tag{14}$$

The proof is given in Appendix A.1. In this lemma, we assume $g$ is smooth enough at $\lambda = 0$. For fair comparison with traditional K-hop message passing framework $\sigma(\sum_{j=0}^K \tau_j A^j HW)$, we just test the flexibility with the similar form $\sigma(\Psi_s HW)$. In this case, we derive the depth $m_\Psi$ necessary for this wavelet basis $\Psi_s$ to induce the amount of mixing $\text{mix}_{y_G}(a, b)$ between two nodes $a$ and $b$.

**Theorem A.2** (**The least depth for mixing**). *Given commute time* $\tau(a, b)$ *(Lovász, 1993) and number of edges* $|E|$. *If* $\Psi_s$ *generates mixing* $\text{mix}_{y_G}(b, a)$, *then the number of layers* $m_\Psi$ *satisfies*

$$m_\Psi \geq \frac{\tau(a,b)}{2K} + \frac{2|E|}{K\sqrt{d_a d_b}}\left[\frac{\text{mix}_{y_G}(b, a)}{\gamma(\alpha^2 s^{2K})^{m_\Psi}} - \frac{1}{\lambda_1}(\gamma + |1 - \lambda^*|^{Km_\Psi+1})\right], \tag{15}$$

*where* $d_a$ *and* $d_b$ *are degrees of two nodes,* $\gamma = \sqrt{\frac{d_{max}}{d_{min}}}$, *and* $|1 - \lambda^*| = \max_{0 < n \leq N-1}|1 - \lambda_n| < 1$.

The proof is given in Appendix A.2. In the following subsections, we firstly prove these lemma and theorem, and finally give the complete proof of Theorem 4.2.

## A.1. Proof of Lemma A.1 (Upper bound for graph wavelet)

*Proof.* We aim to investigate the properties of filters $\Psi_{s_j} = Ug(s_j\lambda)U^\top$ to capture both global and local information, corresponding to the cases $s_j \to 0$ and $s_j \to \infty$, respectively. In the former case, as $s_j$ approaches zero, $g(s_j\lambda)$ tends towards $g(0)$. For the latter case, the spectral information becomes densely distributed and concentrated near zero. Hence, the meaningful analysis of $g(\lambda)$ primarily revolves around $\lambda = 0$. Expanding $g(\lambda)$ using Taylor's series around $\lambda = 0$, we get:

$$g(\lambda) = \sum_{k=0}^K C_k \frac{\lambda^k}{k!} + g^{(K+1)}(\lambda^*)\frac{\lambda^{K+1}}{(K+1)!} \approx \sum_{k=0}^K C_k \frac{\lambda^k}{k!}, \tag{16}$$

where we neglect the high-order remainder term. Next, we have

$$(\Psi)_{ij} = \left(Ug(\Lambda)U^T\right)_{ij} = \left(\sum_{k=0}^K C_k \frac{\hat{\mathcal{L}}^k}{k!}\right)_{ij}$$

$$= \left(\sum_{k=0}^K \frac{C_k}{k!}(I - \hat{A})^k\right)_{ij} = \left(\sum_{k=0}^K \frac{C_k}{k!}\sum_{p=0}^k \binom{k}{p}(-\hat{A})^p\right)_{ij}$$

$$< \left(\sum_{k=0}^K \frac{C_k}{k!}\sum_{p=0}^k \binom{k}{p}(\hat{A})^p\right)_{ij} = \left(\sum_{k=0}^K \frac{C_k}{k!}\sum_{p=0}^k \frac{k!}{(k-p)!p!}(\hat{A})^p\right)_{ij}$$

$$= \left(\sum_{k=0}^K C_k \sum_{p=0}^k \frac{(\hat{A})^p}{(k-p)!p!}\right)_{ij}. \tag{17a}$$

We introduce a new parameter $\mu = \frac{\left(\sum_{k=0}^{K-1} C_k \sum_{p=0}^k \frac{(\hat{A})^p}{(k-p)!p!}\right)_{ij}}{\left(C_K \sum_{p=0}^K \frac{(\hat{A})^p}{(K-p)!p!}\right)_{ij}}$, so the above relation becomes:

$$(\Psi)_{ij} < \left((\mu+1)C_K \sum_{p=0}^K \frac{(\hat{A})^p}{(K-p)!p!}\right)_{ij} = \left(C\sum_{p=0}^K \frac{(\hat{A})^p}{(K-p)!p!}\right)_{ij}, \tag{18}$$

where we set $C = (\mu + 1)C_K$. Then, let us explore the expression $\epsilon_{ij}^p = \frac{(\hat{A})_{ij}^p}{(K-p)!p!}$. First, we will address the denominator $(K - p)!p!$. As $p$ increases, this denominator experiences a sharp decrease followed by a rapid increase. The minimum value occurs at $(K/2)!(K/2)!$ when $p = K/2$, assuming $K$ is even. Second, let's analyze the numerator $(\hat{A})_{ij}^p$, which involves repeated multiplication of $\hat{A}$. According to Theorem 1 in (Li et al., 2018), this repeated multiplication causes $(\hat{A})^p$ to converge to the eigenspaces spanned by the eigenvector $D^{-1/2}\mathbf{1}$ of $\lambda = 0$, where $\mathbf{1} = (1, 1, \ldots, 1) \in \mathbb{R}^{n\,3}$. Then, let us assume there exists a value $p^*$ beyond which the change in $(\hat{A})^p$ becomes negligible. Given that $K$ is a large even number, we can infer that $K/2 \gg p^*$. Thus, when $(K - p)!p!$ sharply decreases, $(\hat{A})^p$ has already approached a stationary state. Consequently, $\max \epsilon_{ij}^p = \frac{(\hat{A})_{ij}^{K/2}}{(K/2)!(K/2)!}$, where the denominator reaches its minimum. Thus, we have

$$
\begin{aligned}
(\Psi)_{ij} &< \left( C \sum_{p=0}^{K} \frac{(\hat{A})^p}{(K-p)!p!} \right)_{ij} \\
&< C(K+1) \left( \frac{(\hat{A})^{K/2}}{(K/2)!(K/2)!} \right)_{ij} \\
&< \left( \frac{C \cdot 2^K (K+1)}{K!} (\hat{A})^{K/2} \right)_{ij}.
\end{aligned}
\tag{19a}
$$

We have $\frac{1}{(K/2)!(K/2)!} < \frac{2^K}{K!}$ given that

$$
\begin{aligned}
(K/2)!(K/2)! &= \left( \frac{K}{2} \cdot \frac{K-2}{2} \cdots \frac{4}{2} \cdot \frac{2}{2} \right) \left( \frac{K}{2} \cdot \frac{K-2}{2} \cdots \frac{4}{2} \cdot \frac{2}{2} \right) \\
&> \left( \frac{K}{2} \cdot \frac{K-2}{2} \cdots \frac{4}{2} \cdot \frac{2}{2} \right) \left( \frac{K-1}{2} \cdot \frac{K-3}{2} \cdots \frac{3}{2} \cdot \frac{1}{2} \right) \\
&= \underbrace{\frac{K \cdot K - 1 \cdot K - 2 \cdot K - 3 \ldots 4 \cdot 3 \cdot 2 \cdot 1}{2 \cdot 2 \cdot 2 \cdot 2 \ldots 2 \cdot 2 \cdot 2 \cdot 2}}_{K \text{ terms}} = \frac{K!}{2^K}.
\end{aligned}
\tag{20}
$$

With $\alpha = \frac{C \cdot 2^K (K+1)}{K!}$ and scale $s$, Eq. (19a) can be finally written as

$$
(\Psi_s)_{ij} < \left( \alpha (\hat{A})^{K/2} s^K \right)_{ij}.
\tag{21}
$$

$\square$

## A.2. Proof of Theorem A.2 (The least depth for mixing)

For this section, we mainly refer to the proof from (Di Giovanni et al., 2023).

**Preliminary.** For simplicity, we follow (Di Giovanni et al., 2023) to denote some operations utilized in this section. As stated, we consider the message passing formula $\sigma(\Psi_s H W)$. First, we denote $h_a^{(l),\alpha}$ as the $\alpha$-th entry of the embedding $h_a^{(l)}$ for node $a$ at the $l$-th layer. Then, we rewrite the formula as:

$$
h_a^{(l),\alpha} = \sigma(\widetilde{h}_a^{(l-1),\alpha}), \quad 1 \le \alpha \le d,
\tag{22}
$$

where $\widetilde{h}_a^{(l-1),\alpha} = (\Psi_s H W)_a$ is the entry $\alpha$ of the pre-activated embedding of node $a$ at layer $l$. Given nodes $a$ and $b$, we denote the following differentiation operations:

$$
\nabla_a h_b^{(l)} := \frac{\partial h_b^{(l)}}{\partial x_a}, \quad \nabla_{ab}^2 h_i^{(l)} := \frac{\partial^2 h_i^{(l)}}{\partial x_a \partial x_b}.
\tag{23}
$$

Next, we firstly derive upper bounds on $\nabla_a h_b^{(l)}$, and then on $\nabla_{ab}^2 h_i^{(l)}$.

---

[3]Simple proof. $(\hat{A})^p = U(I - \Lambda)^p U^\top = \sum_{i=0}^{n} (1 - \lambda_i)^p u_i u_i^\top$. Provided only $1 - \lambda_0 = 1$ and $1 - \lambda_i \in (-1, 1)$ for other eigenvalues, with $p \to \infty$, only $(1 - \lambda_0)^p = 1$ but $(1 - \lambda_i)^p \to 0$. Thus, we have $(\hat{A})^p \to u_1 u_1^\top$, where $u_1 = D^{-1/2}\mathbf{1}$

**Lemma A.3.** *Given the message passing formula $\sigma(\Psi_s HW)$, let assume $|\sigma'| \leq c_\sigma$ and $||W|| \leq w$, where $||\cdot||$ is the operator norm. For two nodes $a$ and $b$ after $l$ layers of message passing, the following holds:*

$$||\nabla_a \boldsymbol{h}_b^{(l)}|| \leq (c_\sigma w)^l (\boldsymbol{B}^l)_{ba}, \tag{24}$$

*where $\boldsymbol{B}_{ba} = \left( \alpha(\hat{\boldsymbol{A}})^{K/2} s^K \right)_{ba}$.*

*Proof.* If $l = 1$ and we fix entries $1 \leq \alpha, \beta \leq d$, then we have:

$$(\nabla_a \boldsymbol{h}_b^{(1)})_{\alpha\beta} = (\text{diag}(\sigma'(\widetilde{\boldsymbol{h}}_b^{(0)}))(\boldsymbol{W}^{(1)} \Psi_{ba} \boldsymbol{I}))_{\alpha\beta}. \tag{25}$$

With Cauchy–Schwarz inequality, we bound the left hand side by

$$||\nabla_a \boldsymbol{h}_b^{(1)}|| \leq ||\text{diag}(\sigma'(\widetilde{\boldsymbol{h}}_b^{(0)}))|| \cdot ||\boldsymbol{W}^{(1)} \Psi_{ba}||$$
$$\leq c_\sigma w \boldsymbol{B}_{ba}.$$

Next, we turn to a general case where $l > 1$:

$$(\nabla_a \boldsymbol{h}_b^{(l)})_{\alpha\beta} = (\text{diag}(\sigma'(\widetilde{\boldsymbol{h}}_b^{(l-1)})(W \sum_j \Psi_{bj} \nabla_a \boldsymbol{h}_j^{(m-1)}))_{\alpha\beta}. \tag{27}$$

Then, we can use the induction step to bound the above equation:

$$||\nabla_a \boldsymbol{h}_b^{(l)}|| \leq (c_\sigma w)^l | \sum_{j_0} \sum_{j_1} \cdots \sum_{j_{l-2}} \Psi_{bj_0} \Psi_{j_0 j_1} \ldots \Psi_{j_{l-3} j_{l-2}} \Psi_{j_{l-2} a}| \tag{28}$$
$$\leq (c_\sigma w)^l (\boldsymbol{B}^l)_{ba}.$$

In Eq. (28), we implicitly use $|\Psi_s^l|_{ba} < \left( \alpha(\hat{\boldsymbol{A}})^{K/2} s^K \right)_{ba}^l = \boldsymbol{B}_{ba}^l$. Similar to proof given in Appendix A.1, we can give the following proof:

$$|\Psi_s^l|_{ba} = |\boldsymbol{U} g(s\Lambda)^l \boldsymbol{U}^T|_{ba} = \left| s^{lK} C^l \frac{\hat{\mathcal{L}}^{lK}}{K!^l} \right|_{ba}$$

$$= \left| s^{lK} \frac{C^l}{K!^l} (\boldsymbol{I} - \hat{\boldsymbol{A}})^{lK} \right|_{ba} = \left| s^{lK} \frac{C^l}{K!^l} \sum_{p=0}^{lK} \binom{lK}{p} (-\hat{\boldsymbol{A}})^p \right|_{ba}$$

$$< \left( s^{lK} \frac{C^l}{K!^l} \sum_{p=0}^{lK} \binom{lK}{p} (\hat{\boldsymbol{A}})^p \right)_{ba} = \left( s^{lK} \frac{C^l}{K!^l} \sum_{p=0}^{lK} \frac{(lK)!}{(lK-p)!p!} (\hat{\boldsymbol{A}})^p \right)_{ba} \tag{29}$$

$$= \left( s^{lK} \frac{C^l (lK)!}{K!^l} \sum_{p=0}^{lK} \frac{(\hat{\boldsymbol{A}})^p}{(lK-p)!p!} \right)_{ba} < \left( s^{lK} \frac{C^l (lK)!}{K!^l} (lK+1) \left( \frac{(\hat{\boldsymbol{A}})^{lK/2}}{(lK/2)!(lK/2)!} \right) \right)_{ba}$$

$$< \left( s^{lK} \frac{C^l (lK)!}{K!^l} (lK+1) \frac{2^{lK}}{(lK)!} (\hat{\boldsymbol{A}})^{lK/2} \right)_{ba} = \left( s^{lK} \frac{C^l \cdot 2^{lK} (lK+1)}{K!^l} (\hat{\boldsymbol{A}})^{lK/2} \right)_{ba}$$

$$< \left( s^{lK} \frac{C^l \cdot 2^{lK} (K+1)^l}{K!^l} (\hat{\boldsymbol{A}})^{lK/2} \right)_{ba} = \left( \alpha(\hat{\boldsymbol{A}})^{K/2} s^K \right)_{ba}^l,$$

where in the last line, we utilize the relation $lK + 1 < (K+1)^l$. $\square$

**Lemma A.4.** *Given the message passing formula $\sigma(\Psi_s HW)$, let assume $|\sigma'|, |\sigma''| \leq c_\sigma$ and $||W|| \leq w$, where $||\cdot||$ is operator norm. For nodes $i$, $a$ and $b$ after $l$ layers of message passing, the following holds:*

$$||\nabla_{ab}^2 \boldsymbol{h}_i^{(l)}|| \leq \sum_{k=0}^{l-1} \sum_{j \in V} (c_\sigma w)^{2l-k-1} w (\boldsymbol{B}^{l-k})_{jb} (\boldsymbol{B}^k)_{ij} (\boldsymbol{B}^{l-k})_{ja}, \tag{30}$$

*where $\boldsymbol{B}_{ba} = \left( \alpha(\hat{\boldsymbol{A}})^{K/2} s^K \right)_{ba}$.*

*Proof.* Considering $\nabla_{ab}^2 \boldsymbol{h}_i^{(l)} \in \mathbb{R}^{d \times (d \times d)}$, we refer to (Di Giovanni et al., 2023) to use the following ordering for indexing the columns:

$$\frac{\partial^2 \boldsymbol{h}_i^{(l),\alpha}}{\partial x_b^\beta \partial x_a^\gamma} := (\nabla_{ab}^2 \boldsymbol{h}_i^{(l)})_{\alpha, d(\beta-1)+\gamma}. \tag{31}$$

Similar to the proof of Lemma A.3, we firstly focus on $m = 1$:

$$(\nabla_{ab}^2 \boldsymbol{h}_i^{(1)})_{\alpha, d(\beta-1)+\gamma} = (\text{diag}(\sigma''(\widetilde{\boldsymbol{h}}_i^{(0),\alpha}))(\boldsymbol{W}^{(1)} \Psi_{ib} \boldsymbol{I})_{\alpha\gamma} \times (\boldsymbol{W}^{(1)} \Psi_{ia} \boldsymbol{I})_{\alpha\beta}. \tag{32}$$

We bound the left-hand side as:

$$||\nabla_{ab}^2 \boldsymbol{h}_i^{(1)}|| \le (c_\sigma w)(w |\boldsymbol{B}_{ib}| |\boldsymbol{B}_{ia}|). \tag{33}$$

Then, for $m > 1$:

$$
\begin{aligned}
&(\nabla_{ab}^2 \boldsymbol{h}_i^{(l)})_{\alpha, d(\beta-1)+\gamma} \\
&= \underbrace{\text{diag}(\sigma''(\widetilde{\boldsymbol{h}}_i^{(l-1),\alpha})(W \sum_j \Psi_{ij} \nabla_a \boldsymbol{h}_j^{(l-1)}) \times (W \sum_j \Psi_{ij} \nabla_b \boldsymbol{h}_j^{(l-1)})}_{\boldsymbol{R}} \\
&+ \underbrace{\text{diag}(\sigma'(\widetilde{\boldsymbol{h}}_i^{(l-1),\alpha})(\boldsymbol{W}^{(m)} \sum_j \Psi_{ij} \nabla_{ab}^2 \boldsymbol{h}_j^{(l-1)})}_{\boldsymbol{Z}}.
\end{aligned}
\tag{34}
$$

We denote $||\nabla_j \boldsymbol{h}_i^{(l-1)}||$ as $(D\boldsymbol{h}^{(l-1)})_{ij}$, and $||\nabla_{ab}^2 \boldsymbol{h}_i^{(l-1)}||$ as $(D^2 \boldsymbol{h}^{(l-1)}{}_{ba})_i$. To bound $\boldsymbol{R}$, we deduce as follows:

$$
\begin{aligned}
||\boldsymbol{R}|| &\le c_\sigma (w \sum_j \boldsymbol{B}_{ij} ||\nabla_a \boldsymbol{h}_j^{(l-1)}||) \times (w \sum_j \boldsymbol{B}_{ij} ||\nabla_b \boldsymbol{h}_j^{(l-1)}||) \\
&= c_\sigma w (w \boldsymbol{B} D \boldsymbol{h}^{(l-1)})_{ib} (\boldsymbol{B} D \boldsymbol{h}^{(l-1)})_{ia} \\
&\le c_\sigma w (w \boldsymbol{B} (c_\sigma w)^{l-1} \boldsymbol{B}^{l-1})_{ib} (\boldsymbol{B} (c_\sigma w)^{l-1} \boldsymbol{B}^{l-1})_{ia} \\
&= (c_\sigma w)^{2l-1} (w (\boldsymbol{B}^l)_{ib} (\boldsymbol{B}^l)_{ia}),
\end{aligned}
\tag{35a}
$$

where we utilize the conclusion from Theorem A.3 in (35a). For term $\boldsymbol{Z}$, we have:

$$
\begin{aligned}
||\boldsymbol{Z}|| &\le c_\sigma w (\boldsymbol{B} D^2 \boldsymbol{h}^{(l-1)})_i \\
&\le c_\sigma w \sum_s \boldsymbol{B}_{is} \sum_{k=0}^{l-2} \sum_{j \in V} (c_\sigma w)^{2l-2-k-1} w (\boldsymbol{B}^{l-1-k})_{jb} (\boldsymbol{B}^k)_{sj} (\boldsymbol{B}^{l-1-k})_{ja} \\
&= \sum_{k=0}^{l-2} \sum_{j \in V} (c_\sigma w)^{2l-2-k} (\boldsymbol{B}^{l-1-k})_{jb} (\boldsymbol{B}^{k+1})_{ij} (\boldsymbol{B}^{l-1-k})_{ja} \\
&= \sum_{k=1}^{l-1} \sum_{j \in V} (c_\sigma w)^{2l-1-k} (\boldsymbol{B}^{l-k})_{jb} (\boldsymbol{B}^k)_{ij} (\boldsymbol{B}^{l-k})_{ja},
\end{aligned}
\tag{36a}
$$

where in (36a), we recursively use the Eq. (34). Finally, we finish the proof as:

$$
\begin{aligned}
||\nabla_{ab}^2 \boldsymbol{h}_i^{(l)}|| &\le ||\boldsymbol{R}|| + ||\boldsymbol{Z}|| \\
&\le \sum_{k=0}^{l-1} \sum_{j \in V} (c_\sigma w)^{2l-1-k} (\boldsymbol{B}^{l-k})_{jb} (\boldsymbol{B}^k)_{ij} (\boldsymbol{B}^{l-k})_{ja}.
\end{aligned}
\tag{37}
$$

$\square$

With Lemma A.3 and A.4, now we give the following theorem.

**Theorem A.5.** *Consider the message passing formula $\sigma(\Psi_s HW)$ with $m_\Psi$ layers, the induced mixing $mix_{y_G}(b, a)$ over the features of nodes $a$ and $b$ satisfies:*

$$mix_{y_G}(b, a) \leq \sum_{l=0}^{m_\Psi - 1} (c_\sigma w)^{(2m_\Psi - l - 1)} \left( w \left( \boldsymbol{B}^{m_\Psi - l} \right)^\top diag \left( \mathbf{1}^\top \boldsymbol{B}^l \right) \boldsymbol{B}^{m_\Psi - l} \right)_{ab}, \tag{38}$$

*where $\boldsymbol{B}_{ba} = \left( \alpha(\hat{\boldsymbol{A}})^{K/2} s^K \right)_{ba}$ and $\mathbf{1} \in \mathbb{R}^n$ is the vector of ones.*

*Proof.* Here, we define the prediction function $y_G : N \times d \to d$ on $G$ as $y_G^{(m_\Psi)} = \texttt{Readout}(\boldsymbol{H}^{(m_\Psi)}\boldsymbol{\theta})$, where $\texttt{Readout}$ is to gather all nodes embeddings to get the final graph embedding, $\boldsymbol{H}^{(m_\Psi)}$ is the node embedding matrix after $m_\Psi$ layers and $\boldsymbol{\theta}$ is the learnable weight for graph-level task. If we set $\texttt{Readout} = \texttt{sum}$, we derive:

$$
\begin{aligned}
\text{mix}_{y_G}(b, a) &= \max_x \max_{1 \leq \beta, \gamma \leq d} \left| \frac{\partial^2 y_G^{(m_\Psi)}(\boldsymbol{X})}{\partial \boldsymbol{x}_a^\beta \partial \boldsymbol{x}_b^\gamma} \right| \\
&\leq \sum_{i \in V} \left| \sum_{\alpha=1}^d \theta_\alpha \frac{\partial^2 h_i^{(m_\Psi), \alpha}}{\partial \boldsymbol{x}_a^\beta \partial \boldsymbol{x}_b^\gamma} \right| \\
&= \sum_{i \in V} \| (\nabla_{ab}^2 \boldsymbol{h}_i^{(m_\Psi)})^\top \boldsymbol{\theta} \| \\
&\leq \sum_{i \in V} \| \nabla_{ab}^2 \boldsymbol{h}_i^{(m_\Psi)} \| \tag{39a} \\
&\leq \sum_{k=0}^{m_\Psi - 1} (c_\sigma w)^{(2m_\Psi - k - 1)} \left( w \left( \boldsymbol{B}^{m_\Psi - k} \right)^\top diag \left( \mathbf{1}^\top \boldsymbol{B}^k \right) \boldsymbol{B}^{m_\Psi - k} \right)_{ab}, \tag{39b}
\end{aligned}
$$

$\square$

where in (39a), we assume the norm $\|\boldsymbol{\theta}\| \leq 1$. In (39b), we use the results from Lemma A.4. This upper bound still holds if $\texttt{Readout}$ is chosen as $\texttt{MEAN}$ or $\texttt{MAX}$ (Di Giovanni et al., 2023).

In theorem A.5, we can assume that $c_\sigma$ to be smaller or equal than one, which is satisfied by the majority of current active functions. Furthermore, considering the normalization (e.g., $L_2$ norm) on $W$, we assume $w < 1$. With these two assumptions, the conclusion of theorem A.5 is rewritten as:

$$\text{mix}_{y_G}(b, a) \leq \sum_{l=0}^{m_\Psi - 1} \left( \left( \boldsymbol{B}^{m_\Psi - l} \right)^\top \text{diag} \left( \mathbf{1}^\top \boldsymbol{B}^l \right) \boldsymbol{B}^{m_\Psi - l} \right)_{ab}. \tag{40}$$

With this new conclusion, we now turn to the proof of Theorem A.2:

*Proof.* Firstly, $\text{diag} \left( \mathbf{1}^\top \boldsymbol{B}^l \right)_i = (\alpha s^K)^l (((\hat{\boldsymbol{A}})^{K/2})^l \mathbf{1})_i \leq \gamma (\alpha s^K)^l$ by using $(((\hat{\boldsymbol{A}})^{K/2})^l \mathbf{1})_i \leq \gamma$ (Di Giovanni et al., 2023). Then, we find

$$
\begin{aligned}
\sum_{l=0}^{m_\Psi - 1} \left( \left( \boldsymbol{B}^{m_\Psi - l} \right)^\top \text{diag} \left( \mathbf{1}^\top \boldsymbol{B}^l \right) \boldsymbol{B}^{m_\Psi - l} \right)_{ab} &\leq \gamma \left( \sum_{l=0}^{m_\Psi - 1} \boldsymbol{B}^{2(m_\Psi - l)} \cdot (\alpha s^K)^l \right)_{ab} \\
&< \gamma \left( \sum_{l=0}^{m_\Psi - 1} (\alpha(\hat{\boldsymbol{A}})^{K/2} s^K)^{2(m_\Psi - l)} \cdot (\alpha s^K)^l \right)_{ab} \\
&< \gamma (\alpha s^K)^{2m_\Psi} \left( \sum_{l=0}^{m_\Psi - 1} \hat{\boldsymbol{A}}^{K(m_\Psi - l)} \right)_{ab} \\
&= \gamma (\alpha s^K)^{2m_\Psi} \left( \sum_{l=1}^{m_\Psi} \hat{\boldsymbol{A}}^{Kl} \right)_{ab}.
\end{aligned} \tag{41}
$$

The following proof depends on *commute time* $\tau(a,b)$ (Lovász, 1993), whose the definition is as follows using the spectral representation of the graph Laplacian (Di Giovanni et al., 2023):

$$\tau(a,b) = 2|E| \sum_{n=0}^{N-1} \frac{1}{\lambda_n} \left( \frac{u_n(a)}{\sqrt{d_a}} - \frac{u_n(b)}{\sqrt{d_b}} \right)^2. \tag{42}$$

Then, we have:

$$\left( \sum_{l=1}^{m_\Psi} \hat{A}^{Kl} \right)_{ab} \leq \sum_{l=0}^{Km_\Psi} \left( \hat{A}^l \right)_{ab}$$

$$= \sum_{l=0}^{Km_\Psi} \sum_{n \geq 0} (1-\lambda_n)^l u_n(a) u_n(b)$$

$$= (Km_\Psi + 1) \frac{\sqrt{d_a d_b}}{2|E|} + \sum_{n>0} \frac{1-(1-\lambda)^{Km_\Psi+1}}{\lambda_n} u_n(a) u_n(b) \tag{43a}$$

$$= (Km_\Psi + 1) \frac{\sqrt{d_a d_b}}{2|E|} + \sum_{n>0} \frac{1}{\lambda_n} u_n(a) u_n(b) - \sum_{n>0} \frac{(1-\lambda)^{Km_\Psi+1}}{\lambda_n} u_n(a) u_n(b).$$

In Eq. (43a), we use $u_0(a) = \sqrt{\frac{d_a}{2|E|}}$. Then, from the definition of commute time, we can get:

$$\sum_{n=1}^{N-1} \frac{1}{\lambda_n} u_n(a) u_n(b) = \frac{-\tau(a,b)}{4|E|} \sqrt{d_a d_b} + \frac{1}{2} \sum_{n>0} \frac{1}{\lambda_n} \left( u_n^2(a) \sqrt{\frac{d_b}{d_a}} + u_n^2(b) \sqrt{\frac{d_a}{d_b}} \right)$$

$$\leq \frac{-\tau(a,b)}{4|E|} \sqrt{d_a d_b} + \frac{1}{2\lambda_1} \left( \sqrt{\frac{d_a}{d_b}} + \sqrt{\frac{d_b}{d_a}} - \frac{\sqrt{d_a d_b}}{|E|} \right), \tag{44}$$

where in the last inequation, we utilize the fact that $\sum_{n>0} u_n^2(a) = 1 - u_0^2(a)$ because $\{u_n\}$ is a set of orthonormal basis. Besides, we use $\lambda_n > \lambda_1, \forall n > 1$. Next, we derive

$$-\sum_{n>0} \frac{(1-\lambda)^{Km_\Psi+1}}{\lambda_n} u_n(a) u_n(b) \leq \sum_{n>0} \frac{|1-\lambda^*|^{Km_\Psi+1}}{\lambda_n} |u_n(a) u_n(b)||$$

$$\leq \frac{|1-\lambda^*|^{Km_\Psi+1}}{2\lambda_1} \sum_{n>0} (u_n^2(a) + u_n^2(b)) \tag{45}$$

$$\leq \frac{|1-\lambda^*|^{Km_\Psi+1}}{2\lambda_1} \left( 2 - \frac{d_a + d_b}{2|E|} \right),$$

where $|1-\lambda^*| = \max_{0<n \leq N-1} |1-\lambda_n| < 1$. Insert derivations (44) and (45) into (43), then gather all above derivations:

$$\text{mix}_{y_G}(b,a) \leq \gamma(\alpha s^K)^{2m_\Psi} \left\{ (Km_\Psi + 1) \frac{\sqrt{d_a d_b}}{2|E|} - \frac{\tau(a,b)}{4|E|} \sqrt{d_a d_b} \right.$$

$$\left. + \frac{1}{2\lambda_1} \left( \sqrt{\frac{d_a}{d_b}} + \sqrt{\frac{d_b}{d_a}} - \frac{\sqrt{d_a d_b}}{|E|} \right) + \frac{|1-\lambda^*|^{Km_\Psi+1}}{2\lambda_1} \left( 2 - \frac{d_a + d_b}{2|E|} \right) \right\} \tag{46}$$

$$\leq \gamma(\alpha s^K)^{2m_\Psi} \sqrt{d_a d_b} \left( \frac{Km_\Psi}{2|E|} - \frac{\tau(a,b)}{4|E|} \right) + \frac{\gamma(\alpha s^K)^{2m_\Psi}}{2\lambda_1} \left( \sqrt{\frac{d_a}{d_b}} + \sqrt{\frac{d_b}{d_a}} \right) + \frac{\gamma(\alpha s^K)^{2m_\Psi}}{\lambda_1} |1-\lambda^*|^{Km_\Psi+1}.$$

In last inequation, we discard $\frac{\sqrt{d_a d_b}}{2|E|} \left[ 1 - \frac{1}{\lambda_1} \left( 1 + \frac{|1-\lambda^*|^{Km_\Psi+1}}{2} \left( \sqrt{\frac{d_a}{d_b}} + \sqrt{\frac{d_b}{d_a}} \right) \right) \right] < 0$ because $\lambda_1 < 1$. Then,

$$\frac{\text{mix}_{y_G}(b,a)}{\gamma(\alpha s^K)^{2m_\Psi} \sqrt{d_a d_b}} \leq \frac{Km_\Psi}{2|E|} - \frac{\tau(a,b)}{4|E|} + \frac{1}{2\lambda_1 \sqrt{d_a d_b}} \left( \sqrt{\frac{d_a}{d_b}} + \sqrt{\frac{d_b}{d_a}} + 2|1-\lambda^*|^{Km_\Psi+1} \right). \tag{47}$$

From (47), we can finally give the lower bound of $m_\Psi$ as:

$$
\begin{aligned}
m_\Psi &\geq \frac{2|E|}{K}\left\{\frac{\tau(a,b)}{4|E|} + \frac{\mathrm{mix}_{y_G}(b,a)}{\gamma(\alpha s^K)^{2m_\Psi}\sqrt{d_a d_b}} - \frac{1}{2\lambda_1\sqrt{d_a d_b}}\left(\sqrt{\frac{d_a}{d_b}} + \sqrt{\frac{d_b}{d_a}} + 2|1-\lambda^*|^{Km_\Psi+1}\right)\right\} \\
&> \frac{2|E|}{K}\left\{\frac{\tau(a,b)}{4|E|} + \frac{1}{\sqrt{d_a d_b}}\left[\frac{\mathrm{mix}_{y_G}(b,a)}{\gamma(\alpha s^K)^{2m_\Psi}} - \frac{1}{2\lambda_1}\left(2\gamma + 2|1-\lambda^*|^{Km_\Psi+1}\right)\right]\right\} \\
&= \frac{2|E|}{K}\left\{\frac{\tau(a,b)}{4|E|} + \frac{1}{\sqrt{d_a d_b}}\left[\frac{\mathrm{mix}_{y_G}(b,a)}{\gamma(\alpha^2 s^{2K})^{m_\Psi}} - \frac{1}{\lambda_1}\left(\gamma + |1-\lambda^*|^{Km_\Psi+1}\right)\right]\right\} \\
&= \frac{\tau(a,b)}{2K} + \frac{2|E|}{K\sqrt{d_a d_b}}\left[\frac{\mathrm{mix}_{y_G}(b,a)}{\gamma(\alpha^2 s^{2K})^{m_\Psi}} - \frac{1}{\lambda_1}\left(\gamma + |1-\lambda^*|^{Km_\Psi+1}\right)\right]
\end{aligned}
\tag{48}
$$

$\square$

### A.3. Proof of Theorem 4.2 (Short-range and long-range receptive fields)

*Proof.* From theorem A.2, we denote $L_{m_\Psi} = \frac{\tau(a,b)}{2K} + \frac{2|E|}{K\sqrt{d_a d_b}}\left[\frac{\mathrm{mix}_{y_G}(b,a)}{\gamma(\alpha^2 s^{2K})^{m_\Psi}} - \frac{1}{\lambda_1}\left(\gamma + |1-\lambda^*|^{Km_\Psi+1}\right)\right]$. For K-order message passing $\sigma(\sum_{j=0}^{K}\tau_j A^j HW)$, $\tau_j \in [0,1]$, we assume that $(\tau_P A^P)_{ba}$ is the maximum among $\{(\tau_0 A^0)_{ba}, \ldots, (\tau_K A^K)_{ba}\}$. According to theorem A.5, we can get the similar conclusion, replacing $B$ with $C = (K+1)\tau_P A^P$. Then, we have the following proof:

*Proof.* Again, $\mathrm{diag}\left(\mathbf{1}^\top C^l\right)_i = ((K+1)\tau_P)^l (A^{Pl}\mathbf{1})_i \leq \gamma((K+1)\tau_P)^l$. Then, we have

$$
\begin{aligned}
\sum_{l=0}^{m_A-1}\left(\left(C^{m_A-l}\right)^\top \mathrm{diag}\left(\mathbf{1}^\top C^l\right) C^{m_A-l}\right)_{ab} &\leq \gamma\left(\sum_{l=0}^{m_A-1} C^{2(m_A-l)}\cdot((K+1)\tau_P)^l\right)_{ab} \\
&< \gamma\left(\sum_{l=0}^{m_A-1}((K+1)\tau_P A^P)^{2(m_A-l)}\cdot((K+1)\tau_P)^l\right)_{ab} \\
&< \gamma((K+1)\tau_P)^{2m_A}\left(\sum_{l=0}^{m_A-1}\hat{A}^{2P(m_A-l)}\right)_{ab} \\
&= \gamma((K+1)\tau_P)^{2m_A}\left(\sum_{l=1}^{m_A}\hat{A}^{2Pl}\right)_{ab} \\
&< \gamma(\sqrt{(K+1)\tau_P})^{4m_A}\left(\sum_{l=1}^{2m_A}\hat{A}^{Pl}\right)_{ab}.
\end{aligned}
\tag{49}
$$

$\square$

Following the rest proof of $L_{m_\Psi}$, replace $\{\alpha s^K, m_\Psi, K\}$ with $\{\sqrt{(K+1)\tau_P}, 2m_A, P\}$, and get the expression of $L_{m_A}$:

$$
L_{m_A} = \frac{\tau(a,b)}{2P} + \frac{2|E|}{P\sqrt{d_a d_b}}\left[\frac{\mathrm{mix}_{y_G}(b,a)}{\gamma((K+1)^2\tau_P^2)^{m_A}} - \frac{1}{\lambda_1}\left(\gamma + |1-\lambda^*|^{2Pm_A+1}\right)\right].
\tag{50}
$$

Therefore, we have

$$
L_{m_\Psi} \approx \frac{P}{K}L_{m_A} + \frac{2|E|}{K\sqrt{d_a d_b}}\left[\frac{\mathrm{mix}_{y_G}(b,a)}{\gamma}\left(\frac{1}{(\alpha^2 s^{2K})^{m_\Psi}} - \frac{1}{((K+1)^2\tau_P^2)^{m_A}}\right)\right],
\tag{51}
$$

where we ignore $|1-\lambda^*|^{Km_\Psi+1}$ and $|1-\lambda^*|^{2Pm_A+1}$. Since $|1-\lambda^*| < 1$ as shown in theorem A.2, therefore $|1-\lambda^*|^{Km_\Psi+1} - |1-\lambda^*|^{2Pm_A+1}$ will be very small, especially when $m_\Psi$ and $m_A$ are large. From Eq. (51), when $s \to \infty$, the relation becomes:

$$
L_{m_\Psi} \approx \frac{P}{K}L_{m_A} - \frac{2|E|}{K(K+1)^{2m_A}\tau_P^{2m_A}\sqrt{d_a d_b}}\frac{\mathrm{mix}_{y_G}(b,a)}{\gamma}.
\tag{52}
$$

Or, when $s \to 0$, the relation becomes:

$$L_{m_\Psi} \approx \frac{P}{K} L_{m_A} + \frac{2|E|}{K\sqrt{d_a d_b}} \frac{\mathrm{mix}_{y_G}(b,a)}{\gamma} \cdot \frac{1}{(\alpha^2 s^{2K})^{m_\Psi}}. \tag{53}$$

$\square$

## B. Details of Encoding Eigenvalues

In this paper, we adopt Eigenvalue Encoding (EE) Module (Bo et al., 2023) to encode eigenvalues. EE functions as a set-to-set spectral filter, enabling interactions between eigenvalues. In EE, both magnitudes and relative differences of all eigenvalues are leveraged. Specifically, the authors use an eigenvalue encoding function to transform each $\lambda$ from scalar $\mathbb{R}^1$ to a vector $\mathbb{R}^d$:

$$\rho(\lambda, 2i) = \sin\left(\epsilon\lambda/10000^{2i/d}\right), \quad \rho(\lambda, 2i+1) = \cos\left(\epsilon\lambda/10000^{2i/d}\right), \tag{54}$$

where $i$ is the dimension of the representations and $\epsilon$ is a hyper parameter. By encoding in this way, relative frequency shifts between eigenvalues are captured. Then, the raw representations of eigenvalues are the concatenation between eigenvalues and corresponding representation vectors:

$$\boldsymbol{Z}_\lambda = [\lambda_1||\rho(\lambda_1), \ldots, \lambda_{N-1}||\rho(\lambda_{N-1})]^\top \in \mathbb{R}^{N \times d}. \tag{55}$$

To capture the dependencies between eigenvalues, a standard Transformer is used followed by skip-connection and feed forward network (FFN):

$$\hat{\boldsymbol{Z}}_\lambda = \mathrm{Transformer}(\mathrm{LN}(\boldsymbol{Z}_\lambda)) + \boldsymbol{Z}_\lambda \in \mathbb{R}^{N \times d}, \quad \boldsymbol{Z} = \mathrm{FFN}(\mathrm{LN}(\hat{\boldsymbol{Z}}_\lambda)) + \hat{\boldsymbol{Z}}_\lambda \in \mathbb{R}^{N \times d}, \tag{56}$$

where LN is the layer normalization. Then, $\boldsymbol{Z}$ is the embedding matrix for eigenvalues, which is injected into the learning of combination coefficients $\tilde{a}$ and $\tilde{b}$, and scales $\tilde{s}$.

## C. Experimental Details

### C.1. Implementation Details

Inspired by (Rampásek et al., 2022), we adopt the hybrid network architecture as shown in Fig. 3, where the "WaveGC" block is the process shown in Fig. 1 (a). This architecture explicitly involves a parallel massage passing neural network (MPNN) (e.g., GCN (Kipf & Welling, 2017), GatedGCN (Bresson & Laurent, 2017)) to augment the low-frequency modeling. Then, these two branches separately go through skip-connection and normalization, and then sum together followed by a two-layers MLP, eventually skip-connection and normalization.

We explore the number of truncated terms $\rho$ from 1 to 10 and adjust the number of scales $J$ from 1 to 5. Additionally, for the pre-defined vector $\overline{s}$ controlling the amplitudes of scales, we test each element in $\overline{s}$ from 0.1 to 10. The usage of the tight frames constraint is also a parameter subject to tuning, contingent on the given dataset. Typically, models iterate through several layers to produce a single result, thus the parameters of WaveGC may or may not be shared between different layers. For short-range datasets, we only retain the

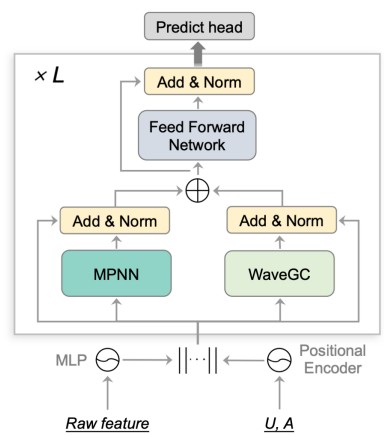

*Figure 3.* Combing MPNN with WaveGC.

first 30% of eigenvalues and their corresponding eigenvectors for efficient eigendecomposition, and set a threshold $\aleph$ and filter out entries in $\Phi$ and $\Psi_{s_j}$ whose absolute value is lower than $\aleph$.

For fair comparisons, we randomly run 4 times on long-range datasets (Dwivedi et al., 2022), and 10 times on short-range datasets (Chen et al., 2022), and report the average results with their standard deviation for all methods. For the sake of reproducibility, we also report the related parameters in Appendix C.7.

_Table 8._ The statistics of the short-range datasets.

| Dataset | # Graphs | # Nodes | # Edges | # Features | # Classes |
|---------|----------|---------|---------|------------|-----------|
| CS | 1 | 18,333 | 163,788 | 6,805 | 15 |
| Photo | 1 | 7,650 | 238,163 | 745 | 8 |
| Computer | 1 | 13,752 | 491,722 | 767 | 10 |
| CoraFull | 1 | 19,793 | 126,842 | 8,710 | 70 |
| ogbn-arxiv | 1 | 169,343 | 1,116,243 | 128 | 40 |

### C.2. Datasets Description

For short-range datasets, we choose five commonly used CS, Photo, Computer, CoraFull (Fey & Lenssen, 2019) and ogbn-arxiv (Hu et al., 2020). CS is a network based on co-authorship, with nodes representing authors and edges symbolizing collaboration between them. In the Photo and Computer networks, nodes stand for items, and edges suggest that the connected items are often purchased together, forming co-purchase networks. CoraFull is a network focused on citations, where nodes are papers and edges indicate citation connections between them. ogbn-arxiv is a citation network among all Computer Science (CS) Arxiv papers, where each node corresponds to an Arxiv paper, and the edges indicate the citations between papers. The details of these five datasets are summarized in Table 8.

_Table 9._ The statistics of the long-range datasets.

| Dataset | # Graphs | Avg. # nodes | Avg. # edges | Prediction level | Task | Metric |
|---------|----------|--------------|--------------|------------------|------|--------|
| PascalVOC-SP | 11,355 | 479.4 | 2,710.5 | inductive node | 21-class classif. | F1 score |
| PCQM-Contact | 529,434 | 30.1 | 61.0 | inductive link | link ranking | MRR |
| COCO-SP | 123,286 | 476.9 | 2,693.7 | inductive node | 81-class classif. | F1 score |
| Peptides-func | 15,535 | 150.9 | 307.3 | graph | 10-task classif. | Avg. Precision |
| Peptides-struct | 15,535 | 150.9 | 307.3 | graph | 11-task regression | Mean Abs. Error |

For long-range tasks, we choose five long-range datasets (Dwivedi et al., 2022), including PascalVOC-SP (VOC), PCQM-Contact (PCQM), COCO-SP (COCO), Peptides-func (Pf) and Peptides-struct (Ps). These five datasets are usually used to test the performance on long-range modeling. VOC and COCO datasets are created through SLIC superpixelization of the Pascal VOC and MS COCO image collections. They are both utilized for node classification, where each super-pixel node is categorized into a specific object class. PCQM is developed from PCQM4Mv2 (Hu et al., 2021) and its related 3D molecular structures, focusing on binary link prediction. This involves identifying node pairs that are in 3D contact but distant in the 2D graph. Both Pf and Ps datasets consist of atomic graphs of peptides sourced from SATPdb. In the Peptides-func dataset, the task involves multi-label graph classification into 10 distinct peptide functional classes. Conversely, the Peptides-struct dataset is centered on graph regression to predict 11 different 3D structural properties of peptides. The details of these five datasets are summarized in Table 9.

### C.3. More analyses for section 6.3

In this section, we firstly give a further visualization on short-range dataset and then analyze the impact of the learned scales.

#### C.3.1. VISUALIZATION ON CORAFULL

To give one more example, we provide additional visualization results on the CoraFull dataset. These results are presented in Fig. 4, where the learned scaling functions $h(\lambda)$ and $g(\lambda)$ meet the specified requirements. The four subfigures in Fig. 4(c) illustrate that as the scale $s_j$ increases, the receptive field of the center node expands. This highlights WaveGC's capability to capture both short- and long-range information by adjusting different values of $s_j$. However, one of our strategies for CoraFull involves considering only 30% of eigenvalues as input. Consequently, the full spectrum is truncated, leaving only the remaining 30% parts, as depicted in Fig. 5. We give a deeper insight in the behavior of this truncation from both spectral and spatial perspectives:

- **Spectral perspective**. As shown in Fig. 5, the wavelet function $g(s\lambda)$ retains non-trivial amplitudes within the first 30% domain. While $g(\lambda) \approx 0$, the retained spectral range is sufficiently broad to allow the wavelets to operate

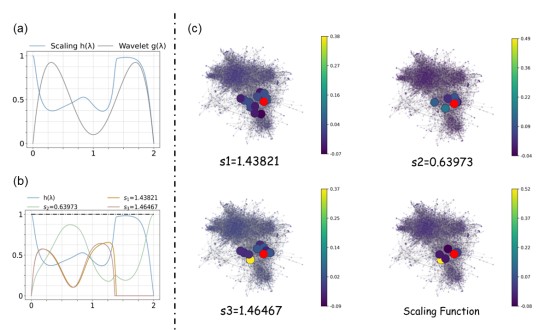
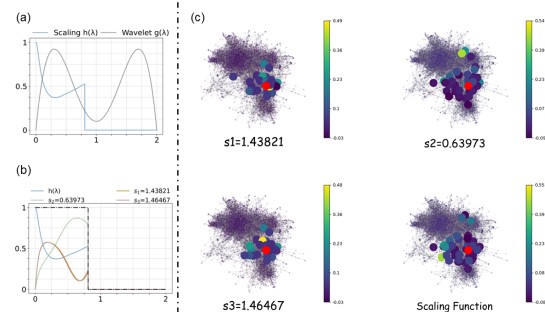

*Figure 4.* Illustration of the spectral and spatial signals of the learned function basis and multiple wavelet bases with full spectrum.

*Figure 5.* Illustration of the spectral and spatial signals of the learned function basis and multiple wavelet bases with partial spectrum.

effectively. Therefore, even in the truncated setting, both the scaling function $h(\lambda)$ and wavelet function $g(\lambda)$ contribute meaningfully to low-frequency modeling.

- **Spatial perspective**. In Fig. 5(b), we observe that truncating the spectrum mimics the effect of using a larger wavelet scale $s > 1$, which reduces the effective spectral range and increases the spatial receptive field. This effect is visually confirmed in Fig. 4(c) and 5(c), where the receptive fields become noticeably larger after truncation. Thus, even on short-range datasets, the wavelet branch captures valuable higher-order information that complements local aggregation from MPNN. This complementary role is further validated by the performance drop observed in Table 6 when wavelets are removed.

Overall, spectral truncation does not impair wavelet behavior; instead, it supports effective low-frequency modeling while also enhancing spatial coverage.

### C.3.2. IMPACT OF THE LEARNED SCALES

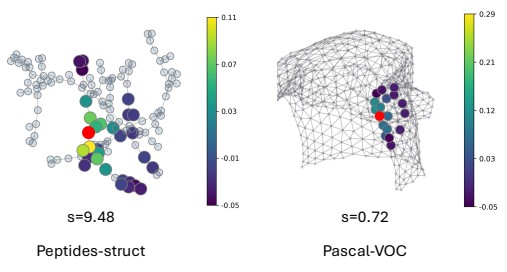

*Figure 6.* Visualizations of receptive fields for Peptides-struct (Ps) and Pascal-VOC (VOC) at their largest scale $s$.

*Table 10.* Comparison of average and max receptive fields of Ps and VOC.

|  | Peptides-struct | Pascal-VOC |
|---|---|---|
| Avg. Receptive Field | 3.02 | 0.74 |
| Max Receptive Field | 9 | 3 |
| Avg. Shortest Path | 20.89 | 10.74 |

We empirically analyze the learned scale values and their impact on receptive fields in Fig. 6. Specifically, we illustrate the largest learned scales for the Ps and VOC datasets, along with their corresponding receptive field visualizations. The receptive field is heuristically defined as follows:

***Definition*** 1. (Receptive field.) Given a wavelet $\Psi(s\lambda)$, node $j$ lies in the receptive field of node $i$ if $|\Psi(s\lambda)[i,j]| > 0.1 \times \max(|\Psi(s\lambda)|)$.

Under this criterion, we observe that Ps exhibits larger receptive fields, corresponding to a larger learned scale of 9.48. We further report the average and maximum receptive field sizes across all nodes in Table 10. The larger receptive fields in Ps align with its inherently longer average shortest-path distances, thus validating the model's ability to adaptively adjust to long-range dependencies. To examine the extreme case of large-scale values, we increase the predefined scale vector $\bar{s}$ in Eq. (7) for Peptides-func (Pf) to (10, 100, 1000). This vector determines the upper bound of the learnable scale range. The

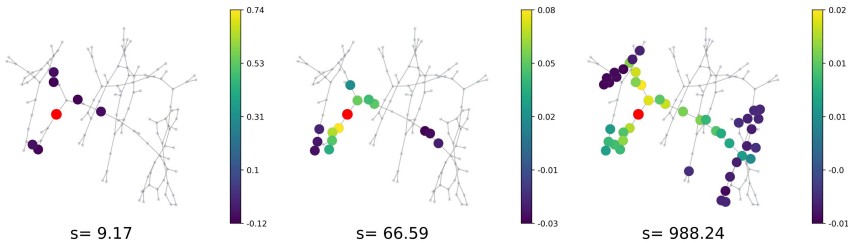

*Figure 7.* Visualizations of receptive fields for Peptides-func (Pf) at extreme scales.

resulting learned scales and receptive fields are depicted in Figure 3. When s = 9.17, the red node primarily aggregates local information; in contrast, at s = 988.24, the same node gathers information from a much broader range. This confirms our theoretical assertion that WaveGC exhibits long-range behavior as s approaches infinity.

## C.4. More comparisons between WaveGC and ChebNet

*Table 11.* More ablations for differences between WaveGC and ChebNet.

|  | Free $\tilde{\alpha}$ | Free $\tilde{\beta}$ | Fix s=1 | Free $\tilde{s}$ | Original |
|---|---|---|---|---|---|
| Computer | 91.28±0.15 | 91.19±0.09 | 91.73±0.02 | 91.51±0.02 | **92.26±0.18** |
| Ps | 25.08±0.01 | 25.09±0.12 | 25.28±0.00 | 25.15±0.25 | **24.83±0.11** |

Obviously, both WaveGC and ChebNet attempt weighted combination of Chebyshev polynomials in different ways. On one hand, ChebNet learns term coefficients independently, while WaveGC map eigenvectors into coefficients $\tilde{\alpha}$ and $\tilde{\beta}$. On the other hand, WaveGC further involve multiple and learnable scales $\tilde{s}$. Finally, we test importance of these differences on the Computer and Ps. The results are summarized in Table 11, showcasing different variants such as free learning coefficients (i.e., $\tilde{\alpha}$, $\tilde{\beta}$), adopting single scale s=1, and free learning $\tilde{s}$ to avoid joint parameterization. Each of these modifications resulted in degraded performance compared to the original model, demonstrating the improvements our new model offers over ChebNet.

## C.5. Complexity and Running time

*Table 12.* Comparison on running time per epoch (s).

|  | SGWT | GWNN | WaveShrink | WaveNet | DEFT | UFGConvS | UFGConvR | WaveGC |
|---|---|---|---|---|---|---|---|---|
| Computer | 3.6 | 0.8 | 2.7 | 2.0 | 1.1 | 3.1 | 3.2 | 1.5 |
| Ps | 21.0 | 30.5 | 52.0 | 27.3 | 23.6 | 47.3 | 43.5 | 23.9 |

The main contribution of WaveGC is to address long-range interactions in graph convolution, so it inevitably establishes spatial connections between distant nodes. This results in the same $O(N^2)$ complexity as Transformer (Vaswani et al., 2017). This is the same for all spectral graph wavelets, including SGWT, GWNN, WaveShrink, WaveNet and UFGConvS/R. A possible solution is to decrease the number of considered frequency modes from $N$ to $\nu$. In this way, the complexity is reduced to $O(\nu \cdot N)$. We report the running time consumption of WaveGC and other spectral graph wavelets (that is, SGWT, GWNN, WaveShrink, WaveNet, DEFT, UFGConvS and UFGConvR). The time consumptions for `Computer` and `Ps` are presented in Table 12. According to the table, the running time of WaveGC is in the first level among spectral graph wavelets.

## C.6. Hyper-Parameter Sensitivity Analysis

In WaveGC, two key hyper-parameters, namely $\rho$ and $J$, play important roles. The parameter $\rho$ governs the number of truncated terms for both $T_i^o$ and $T_i^e$, while $J$ determines the number of scales $s_j$ in Eq. (7). In this section, we explore the sensitivity of $\rho$ and $J$ on the Peptides-struct (Ps) and Computer datasets. The results are visually presented in Fig.8, where the color depth of each point reflects the corresponding performance (the lighter the color, the better the performance), and

the best points are identified with a red star. Observing the results, we note that the optimal value for $\rho$ is 2 for Ps and 7 for Computer. This discrepancy can be attributed to the substantial difference in the graph sizes between the two datasets, with Computer exhibiting a significantly larger graph size (refer to Appendix C.2). Consequently, a more intricate filter design is necessary for the larger dataset. Concerning $J$, the optimal value is determined to be 3 for both Ps and Computer. A too small $J$ leads to inadequate coverage of ranges, while an excessively large $J$ results in redundant scales with overlapping ranges.

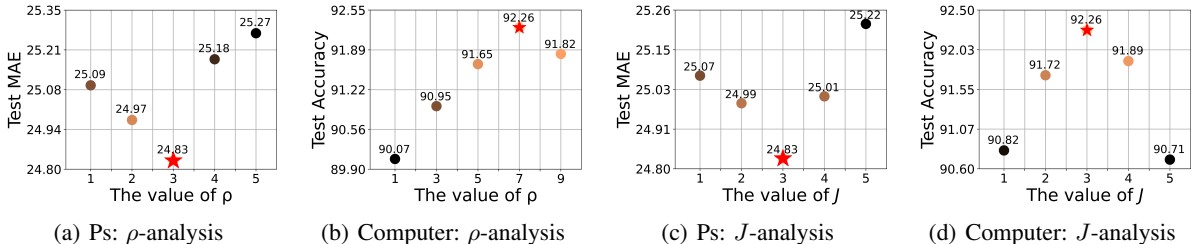

(a) Ps: $\rho$-analysis    (b) Computer: $\rho$-analysis    (c) Ps: $J$-analysis    (d) Computer: $J$-analysis

*Figure 8.* Analysis of the sensitivities of $\rho$ and $J$.

### C.7. Hyper-parameters Settings

We implement our WaveGC in PyTorch, and list some important parameter values in our model in Table 13. Please note that for the five long-range datasets, we follow the parameter budget $\sim$500k (Dwivedi et al., 2022).

*Table 13.* The values of parameters used in WaveGC (T: True; F: False).

| Dataset | # parameters | $\rho$ | $J$ | $\overline{s}$ | Tight frames | $\aleph$ |
|---|---|---|---|---|---|---|
| CS | 495k | 3 | 3 | $\{0.5, 0.5, 0.5\}$ | T | 0.1 |
| Photo | 136k | 3 | 3 | $\{1.0, 1.0, 1.0\}$ | T | 0.1 |
| Computer | 167k | 7 | 3 | $\{10.0, 10.0, 10.0\}$ | T | 0.1 |
| CoraFull | 621k | 3 | 3 | $\{2.0, 2.0, 2.0\}$ | T | 0.1 |
| ogbn-arxiv | 2,354k | 3 | 3 | $\{5.0, 5.0, 5.0\}$ | F | / |
| PascalVOC-SP | 506k | 5 | 3 | $\{0.5, 1.0, 10.0\}$ | T | / |
| PCQM-Contact | 508k | 5 | 3 | $\{0.5, 1.0, 5.0\}$ | T | / |
| COCO-SP | 546k | 3 | 3 | $\{0.5, 1.0, 10.0\}$ | T | / |
| Peptides-func | 496k | 5 | 3 | $\{10.0, 10.0, 10.0\}$ | T | / |
| Peptides-struct | 534k | 3 | 3 | $\{10.0, 10.0, 10.0\}$ | F | / |

### C.8. Operating Environment

The environment where our code runs is shown as follows:

- Operating system: Linux version 5.11.0-43-generic

- CPU information: Intel(R) Xeon(R) Gold 6226R CPU @ 2.90GHz

- GPU information: NVIDIA RTX A5000

## D. Approximation Strategy for O(N) Complexity

To further reduce complexity, we propose a fully polynomial-based approximation that removes the need for eigendecomposition, achieving total complexity of O(N), on par with graph Fourier-basis-based methods. This is achieved via polynomial approximation of the wavelet transform using Chebyshev polynomials:

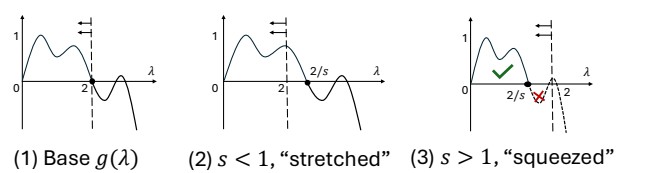

(1) Base $g(\lambda)$    (2) $s < 1$, "stretched"    (3) $s > 1$, "squeezed"        true $g(s\lambda)$     complete $g(s\lambda)$    "window" $w(\lambda)$

*Figure 9.* The scale $s$ can "stretch" or "squeeze" the shape of $g(\lambda)$ as $g(s\lambda)$.

*Figure 10.* The illustration of applying "window" over $g(s\lambda)$.

- **Scaling function** $h(\Lambda)$. Since $h(\boldsymbol{\Lambda}) = \sum b_i T_i^o(\boldsymbol{\Lambda})$, where $T_i^o$ are odd-degree Chebyshev polynomials, we can compute

$$\boldsymbol{\Phi}\boldsymbol{f} = \boldsymbol{U}h(\boldsymbol{\Lambda})\boldsymbol{U}^\top\boldsymbol{f} = \sum b_i T_i^o(\boldsymbol{L})\boldsymbol{f}. \tag{57}$$

This is equivalent to a polynomial operation over the graph Laplacian $\boldsymbol{L}$, which has $O(N)$ complexity.

- **Wavelet Function** $g(\Lambda)$. Similarly, $g(\boldsymbol{\Lambda}) = \sum a_i T_i^e(\boldsymbol{\Lambda})$, where $T_i^e$ are even-degree Chebyshev polynomials, gives

$$\boldsymbol{\Psi}\boldsymbol{f} = \boldsymbol{U}g(\boldsymbol{\Lambda})\boldsymbol{U}^\top\boldsymbol{f} = \sum a_i T_i^e(\boldsymbol{L})\boldsymbol{f}, \tag{58}$$

which is also polynomial in $\boldsymbol{L}$ with $O(N)$ cost.

- **Incorporating scale** $s$. The domain $\lambda \in [0,2]$ for $g(\lambda)$ transforms to $\lambda \in [0, 2/s]$ in $g(s\lambda)$. This raises two scenarios:

  - If $s < 1$: The full spectrum [0,2] is covered, and $g(s\lambda)$ remains valid as a polynomial (Fig. 9 (2)).
  - If $s > 1$: Only the interval [0,2/s] is valid. The rest of the spectrum [2/s,2] should be suppressed (Fig. 9 (3)). To handle this, we apply a *window function* $w(\lambda)$, where:

$$w(\lambda) = \begin{cases} 1 & \lambda \in [0, 2/s] \\ 0 & \lambda \in [2/s, 2] \end{cases}$$

  The true scaled wavelet becomes $g(s\lambda) = g(s\lambda) \cdot w(\lambda)$. Both $g(s\lambda)$ and $w(\lambda)$ can be approximated using Chebyshev polynomials, so the entire operation remains within $O(N)$ complexity.

Using this approach, plus without eigenvalue encoding (EE) and tight frame constraint, we no longer require EVD with the maximum simplification. The resulting model maintains the theoretical structure of WaveGC while gaining substantial computational benefits.

*Table 14.* Running time (s) per epoch.

|  | CS | Photo | Computer |
|---|---|---|---|
| GPRGNN | 1.1 | 0.2 | 0.4 |
| BernNet | 1.5 | 0.5 | 1.3 |
| UniFilter | 5.7 | 0.8 | 1.5 |
| WaveGC_simplified | 1.4 | 0.6 | 1.8 |

*Table 15.* Qualified results on three short-range datasets.

| Accuracy ↑ | CS | Photo | Computer |
|---|---|---|---|
| GPRGNN | 95.13 | 94.49 | 90.82 |
| BernNet | 95.42 | 94.67 | 90.98 |
| UniFilter | 95.68 | 94.34 | 90.07 |
| WaveGC_simplified | 95.63 | 94.90 | 91.22 |
| WaveGC | 95.89 | 95.37 | 92.26 |

To validate this simplified version, we compared its runtime and accuracy with GPRGNN (Chien et al., 2020), BernNet (He et al., 2021), and UniFilter (Huang et al., 2024) on three short-range datasets. As shown in Table 14, the *WaveGC_simplified* achieves comparable training time to Fourier-based methods. According to Table 2, it incurs only a small drop in performance compared with WaveGC, confirming that polynomial approximation remains effective even without EVD.

# E. Related Work

**Graph Wavelet Transform.** Graph wavelet transform is a generalization of classical wavelet transform (Mallat, 1999) into graph domain. SGWT (Hammond et al., 2011) defines the computing paradigm on weighted graph via spectral graph theory. Specifically, it defines scaling operation in time field as the scaling on eigenvalues. The authors also prove the localization properties of SGWT in the spatial domain in the limit of fine scales. To accelerate the computation on transform, they additionally present a fast Chebyshev polynomial approximation algorithm. GWNN (Xu et al., 2019a) chooses heat kernel as the filter to construct the bases. The graph wavelet bases learnt from these methods are not guaranteed as band-pass filters in $\lambda \in [0, 2]$ and thus violate admissibility condition (Mallat, 1999). UFGCONV (Zheng et al., 2021) defines a framelet-based graph convolution with Haar-type filters. WaveNet (Yang et al., 2024) relies on Haar wavelets as bases, and uses the highest-order scaling function to approximate all the other wavelets and scaling functions. WGGP (Opolka et al., 2022) integrates Gaussian processes with Mexican Hat to represent varying levels of smoothness on the graph. The above four methods fix the form of the constructed wavelets, extremely limiting the adaptivity to different datasets. In this paper, our WaveGC constructs band-pass filter and low-pass filter purely depending on the even terms and odd terms of Chebyshev polynomials. In this case, the admissibility condition is strictly guaranteed, and the constructed graph wavelets can be arbitrarily complex and flexible with the number of truncated terms increasing. In addition, SEA-GWNN (Deb et al., 2024) focuses on the second generation of wavelets, or lifting schemes, which is a different topic from ours.

**Graph Scattering Transform.** The Scattering Transform constructs a hierarchical, tree-like structure by combining a cascading filter bank (or wavelets), point-wise non-linearity, and a low-pass operator. As introduced by Mallat (Mallat, 2012), this approach guarantees translation invariance and stability to deformations. On one hand, researchers have explored the application of this technique to graph data. Early efforts, such as those by (Zou & Lerman, 2020), (Gama et al., 2019), and GS-SVM (Gao et al., 2019), extended the scattering transform into the graph spectral domain. ST-GST (Pan et al., 2020) defined filtering and wavelets for spatio-temporal graphs, deriving the corresponding scattering process. Meanwhile,(Gama et al., 2018) employed lazy diffusion(Coifman & Maggioni, 2006) as wavelets to construct graph diffusion scattering, demonstrating its stability against deformations based on diffusion distance. HDS-GNN (Zhang et al., 2022) enhanced GNNs by integrating scattering features from a diffusion scattering network layer by layer, while GGSN (Koke & Kutyniok, 2022) introduced further flexibility to each operation. On the other hand, the computational efficiency of this transform, which involves a total of $\sum_{l=1}^{L} J^l$ filtering operations, poses a significant challenge. To address this, pGST (Ioannidis et al., 2020) proposed a pruning strategy, retaining only the higher-energy child signals for each parent node. Scattering GCN (Min et al., 2020) further optimized the process by selectively using more beneficial wavelets, simplifying the scattering computation.

**Spectral graph convolution.** Traditional studies on spectral graph convolution mainly concentrate on the design of filter with fixed Fourier bases. One way is to design low-pass filters that smooth signals within neighboring regions. GCN (Kipf & Welling, 2017) keeps the first two ChebNet (Defferrard et al., 2016) terms with extra tricks, and averages signals between neighbors. PPNP (Gasteiger et al., 2018) smooths signals in a broader range following PageRank based diffusion. Another way is to design adaptive filters so work in both homophily and heterophily scenarios. ChebNet (Defferrard et al., 2016) approximates universe filter functions with learnable coefficients before each Chebyshev term. FAGCN (Bo et al., 2021) proposes self-gating mechanism to adaptively learn more information beyond low-frequency information in GNNs.

**Graph Transformer.** Graph Transformer (GT) has attracted considerable attentions on long-range interaction. GT (Dwivedi & Bresson, 2020) proposes to employ Laplacian eigenvectors as PE with randomly flipping their signs. Graphormer (Ying et al., 2021) takes the distance of the shortest path between two nodes as spatial encoding, which is involved in attention calculation as a bias. GraphGPS (Rampásek et al., 2022) provides different choices for PE, consisting of LapPE, RWSE, SignNet and EquivStableLapPE. SGFormer (Wu et al., 2023) is empowered by a simple attention model that can efficiently propagate information among arbitrary nodes. Recently, Xing et al. (2024) are the first to reveal the over-globalizing problem in graph transformer, and propose CoBFormer to improve the GT capacity on local modeling with a theoretical guarantee.

