# OpenReview forum: "A General Graph Spectral Wavelet Convolution via Chebyshev Order Decomposition"
_ICML.cc/2025/Conference — ICML 2025 poster_

### Official Review · Reviewer_5Zza · 2025-03-10

**Overall Recommendation:** 3

**Summary:**

The authors propose a novel spectral graph network, WaveGC, inspired by SWGT. WaveGC filters input features with matrix-valued kernels in the spectral domain and utilizes transformer architectures for wavelet transforms. The wavelet functions are learnable and are parameterized by Chebyshev polynomials with learnable spatial scales and polynomial coefficients. The authors provide theoretical results on the short-range and long-range performance of WaveGC and demonstrate its effectiveness on both node-level and graph-level tasks.

**Claims And Evidence:**

The authors supported their claims with evidence.

**Essential References Not Discussed:**

While there were concerns about WaveGC missing the prior work (Bastos et al., 2022) that learns spectral wavelets, I see that the authors have included it as a baseline and shown competitive performance against it. I agree the novelty is somewhat limited along this line, but I think it is fine as long as the authors do not overclaim.

**Experimental Designs Or Analyses:**

The experimental design and analyses make sense to me.

**Methods And Evaluation Criteria:**

The methods and evaluation criteria make sense to me.

**Other Comments Or Suggestions:**

Interestingly I am assigned the same paper that I reviewed a year ago. Most of my concerns, e.g. missing evaluation on large datasets and using eigenvalue embeddings, were addressed during the rebuttal period last year and in this new submission. Thus I maintain the same score as last year.

**Other Strengths And Weaknesses:**

Strengths: The authors analyze WaveGC theoretically. The method is evaluated on large datasets. The authors also provide details that rationalize WaveGC in Section 5.
Weaknesses: The quality of Figure 1 is still not good enough.

**Questions For Authors:**

The authors now advertise WaveGC satisfying admissibility criteria as the main advantage against (Bastos et al., 2022). I understand this is theoretically more appealing, but what is the practical advantage of being admissible?

**Relation To Broader Scientific Literature:**

The authors proposed a new spectral GNN architecture based on graph wavelets. Good enough for an ICML submission.

**Theoretical Claims:**

The theoretical results make sense to me.

---

> ### Author Rebuttal · Authors · 2025-03-31
>
> We thank the referee for taking the time to review our paper. Please, see below our answer to the raised comments/questions.
>
> > Q1: The quality of Figure 1 is still not good enough.
>
> We have improved the image resolution of Figure 1 at https://drive.google.com/file/d/1rfuYedo4FkeLg1zClq45ZOHU6rH-p8L4, and will update the figure in the next version of the paper.
>
> > Q2: Practical advantage of being admissible.
>
> |        |     Actor      |    Tolokers    |
> | :---: | :------------: | :------------: |
> |  DEFT  |   35.04±1.89   |   84.32±0.45   |
> | WaveGC | **37.40±1.04** | **85.38±0.52** |
> Table 1. Comparing WaveGC and DEFT on heterophily datasets.
>
> The wavelet admissibility criterion is not just a theoretical preference—it is a fundamental requirement for a function to be considered a valid wavelet. Without satisfying this condition (i.e., g(0)=0), a function labeled as a "wavelet" **loses its essential localization property**. As discussed in our response to Reviewer 4MPc (Q1), this localization enables node-specific filtering, where each node can capture local deviations from the global signal structure. In contrast, non-admissible constructions (e.g., DEFT [Bastos et al., 2022]) lack this guarantee, leading to **less precise node-level modeling**.
>
> The **practical advantage** of admissibility becomes evident in tasks that require strong node-wise representation learning. For instance:
>
> $\bullet$ In **short-range node-level tasks** (e.g., CS, Photo, Computer, CoraFull, ogbn-arxiv), WaveGC consistently outperforms baselines.
>
> $\bullet$  In **long-range node-level tasks** (e.g., VOC and COCO), WaveGC also demonstrates superior performance.
>
> $\bullet$ To further validate the node-level expressiveness, we conducted additional experiments on **heterophilous datasets** (e.g., Actor and Tolokers), where accurate node-level modeling is particularly challenging. Results, shown in Table 1, confirm that WaveGC outperforms DEFT.
>
> Together, these results across **homophily, heterophily, short-range, and long-range settings** support the claim that admissibility leads to more effective and localized node-wise signal processing, providing WaveGC with a substantial practical advantage over prior non-admissible designs.

---

> > ### Comment · Reviewer_5Zza · 2025-04-09
> >
> > I thank the authors for their responses and clarifications. I maintain my positive rating.

---

> > > ### Author Response · Authors · 2025-04-09
> > >
> > > Thank you for your positive evaluation and rating. We will continue to improve the paper based on your and other reviewers' valuable feedback further.

---

### Official Review · Reviewer_4MPc · 2025-03-11

**Overall Recommendation:** 2

**Summary:**

This paper proposes a wavelet-based graph convolution method, WaveGC, which integrates spectral bases and matrix-valued kernels. The authors use the odd and even terms of Chebyshev polynomials to learn graph wavelets that satisfy the necessary conditions. Experimental results demonstrate that the proposed method achieves promising improvements on both short-range and long-range tasks.

**Claims And Evidence:**

The authors discuss the limitations of Fourier bases in Section 3.1, arguing that they lack multi-resolution and adaptability. This serves as the main motivation for exploring graph wavelets in this paper. However, to my knowledge, FFT bases are also fixed, much like graph Fourier bases. Additionally, while graph wavelets are theoretically more flexible than graph Fourier bases, they do not seem to offer significant practical advantages. Notably, the authors' experiments do not provide a comparison with graph Fourier base-based methods, which could have helped to highlight the benefits of wavelet-based methods.

**Essential References Not Discussed:**

There is no lack of essential references, and the authors have considered the key works in the field of graph wavelets.

**Experimental Designs Or Analyses:**

The main issue in the experimental section is the absence of baselines based on Fourier bases, including polynomial approximation methods such as GPR-GNN(https://arxiv.org/abs/2006.07988), BernNet(https://arxiv.org/abs/2106.10994), UniFilter(https://arxiv.org/abs/2405.12474), S²GCN(https://arxiv.org/abs/2405.19121), and others.

**Methods And Evaluation Criteria:**

This paper introduces novel wavelet-based graph convolution methods and proposes using odd and even Chebyshev polynomials to learn graph wavelets. This approach offers a new method for graph convolution techniques and holds some implications for the graph learning community.

**Other Comments Or Suggestions:**

Minor comments:
The $\mathbf{I}_n$ on the left of line 108 should be written as $\mathbf{I}_N$.
The K on the left in line 285 should be in math font.

**Other Strengths And Weaknesses:**

The significant computational complexity of eigenvalue decomposition remains a key drawback, although it is present in most graph wavelet methods. The proposed method in this paper faces challenges in scaling to large graphs. In contrast, graph Fourier-basis-based methods can be easily scaled to graphs with hundreds of millions of nodes.

**Questions For Authors:**

1. Further discussion is needed on how the graph Fourier-basis-based methods compare with the method proposed in this paper, both theoretically and experimentally. This comparison should especially contain some of the latest approaches in the field.

2. The functions $g$ and $h$ approximated by odd and even Chebyshev polynomials are sufficient but not necessary conditions for any function $g$ and $h$ that satisfy the graph wavelet condition. Does this imply that using odd and even Chebyshev polynomials to approximate $g$ and $h$ does not guarantee that any function meeting the requirements will be learned?


## update after rebuttal

I keep my score.

**Relation To Broader Scientific Literature:**

The discussion in this paper is somewhat limited to the narrow scope of graph wavelets. If the authors intend to explore spectral-based graph convolution methods, they should focus on comparing them with existing state-of-the-art methods based on Fourier bases, as this could lead to meaningful advancements in the field of graph learning.

**Theoretical Claims:**

The main theoretical contribution of this paper is Theorem 4.2. I reviewed the corresponding proof and found no issues with it.

---

> ### Author Rebuttal · Authors · 2025-03-31
>
> We thank the reviewer for the careful reading and comments regarding our work. Please, see below our answer to the raised comments/questions. The link for the table is available at https://drive.google.com/file/d/146c3rsB_eJLJmk3I21aWg9fzGPpCvd7S.
>
> > Q1: Compare with SOTA methods based on Fourier bases with further discussion theoretically and experimentally.
>
> Below we address both experimental and theoretical aspects:
> 1. Experimental Comparison
>
> We have conducted additional experiments comparing WaveGC with four representative Fourier basis-based methods—**GPR-GNN, BernNet, UniFilter**, and **S²GCN**—on three long-range and three short-range datasets. As shown in the newly added Figure 1, WaveGC consistently outperforms these baselines, with particularly strong gains on long-range tasks. This empirically supports the practical advantages of our wavelet-based design, especially in modeling multi-scale information.
>
> 2. Theoretical Perspective
>
> a) The core distinction between WaveGC and Fourier-based spectral methods lies in their **design focus**:
>
> $\bullet$ Fourier-based methods typically fix the basis (Laplacian eigenvectors) and focus on designing flexible **filters**.
>
> $\bullet$ WaveGC, by contrast, introduces a novel **learnable basis construction** via graph wavelets and Chebyshev decomposition, while using a matrix-valued kernel as the filter.
>
> These two directions—basis design and filter design—are **complementary** rather than mutually exclusive. In fact, a hybrid method (e.g., applying Bernstein polynomials as filters on our wavelet bases, akin to a "BernWave" model) could be a promising future direction.
>
> b) Localization and Adaptivity
>
> Graph wavelets offer **node-level localization** due to the property g(0)=0 and the small values of g(λ) at low frequencies. This enables each node to respond to local deviations from global structure, creating **node-dependent filters**. In contrast, Fourier bases are inherently global, with eigenvectors representing community-level structures—leading to **community-dependent filters**.
>
> WaveGC benefits from this localized adaptivity, further enhanced by the scaling parameter s, which enables **multi-resolution control**. This is critical for tasks requiring both short- and long-range information modeling.
>
> c) Scope of Fourier Methods
>
> While many recent Fourier-based methods are effective in scenarios such as homophily/heterophily adaptation (e.g., GPR-GNN), they do not explicitly model spatial scale or receptive field. Even methods like S²GCN, which improve long-range capture via kernel design, may be limited by the fixed Fourier basis. Combining such techniques with our wavelet-based approach could potentially lead to stronger long-range generalization.
>
> We will include the new experimental results and expand our discussion in the paper to address these points.
>
> > Q2: Scaling to large graphs.
>
> 1) Different Designs for Different Scenarios: While Fourier-based methods scale well to massive graphs, they often sacrifice expressiveness—particularly in long-range modeling, where WaveGC consistently outperforms them (Table 1 in the link). We believe model design should align with task demands rather than aim for universal scalability.
> 2) Applicability to Current Datasets: The eigendecomposition required for WaveGC has O($N^3$) complexity in the worst case and O($N^2$) in training. This is practical for the small-to-medium graphs used in long-range benchmarks, where detailed spectral modeling is critical.
> 3) Our focus is on constructing **learnable, admissible wavelet bases**—a core challenge in spectral graph learning. Scaling spectral methods efficiently is an important but orthogonal problem that we identify as promising future work.
>
> > Q3: Can WaveGC approximate functions meet the requirements?
>
> We agree that using odd and even Chebyshev polynomials to construct h(λ) and g(λ) provides a **sufficient but not necessary** condition for wavelet admissibility.  While our method does not guarantee representation of all admissible function pairs (g,h), it introduces a **strong inductive bias** through the separation of odd and even Chebyshev terms, enabling a principled and controllable wavelet construction.
>
> Despite this restriction, our approach offers **strong practical approximation capacity** due to several factors:
>
> $\bullet$ Chebyshev polynomials form a complete basis over [0,2], allowing approximation of any continuous function on this interval.
>
> $\bullet$ The separation into odd and even terms allows independent control over h and g while preserving admissibility.
>
> $\bullet$ Learnable coefficients enhance flexibility in shaping the filter functions.
>
> $\bullet$ The scaling parameter s further adapts the spectral response.
>
> Together, these elements form a flexible and expressive class of wavelet filters suitable for practical graph learning. We will clarify this approximation perspective in the revised manuscript.

---

> > ### Comment · Reviewer_4MPc · 2025-04-02
> >
> > Do the authors have any improving methods for dealing with WaveGC's high computational complexity, especially compared with graph Fourier-basis-based methods?

---

> > > ### Author Response · Authors · 2025-04-06
> > >
> > > We appreciate the reviewer's suggestion. We can consider existing strategies as well as propose newly developed approaches to reduce the computational complexity of our model. Please kindly find tables and figures at https://drive.google.com/file/d/1cLakrmKX0nOlcyv_RC2_e38npfxiyKlJ/view?usp=sharing.
> > >
> > > **1\) Existing Efficiency Measures**
> > > Our current implementation incorporates 1\) efficient eigendecomposition with $O(N^2)$ and 2\) retaining the first k eigenvalues with O(kN) for the following computation. This complexity is feasible for the small- to medium-sized graphs in our experiments.
> > >
> > > **2\) New Approximation Strategy for O(N) Complexity**
> > >
> > > To further reduce complexity, we propose a fully polynomial-based approximation that removes the need for eigendecomposition, achieving total complexity of O(N), comparable to Fourier-basis-based methods:
> > >
> > > * **Scaling Function h(λ) and Wavelet Function g(λ):**
> > >
> > >   Given $h(λ)=\sum b_iT_i^o(\lambda)$, the transform becomes $\Phi f=Uh(\Lambda)U^\top f=\sum b_iT_i^o(Lf)$, a Chebyshev polynomial over the Laplacian L. Similarly, $g(λ)=\sum a_iT_i^e(λ)$ leads to $\Psi f=Uh(\Lambda)U^\top f=\sum a_iT_i^e(Lf)$. Both operations are polynomial and thus O(N).
> > >
> > > * **Handling the scale s:**
> > >   The domain λ∈\[0,2\] for g(λ) transforms to λ∈\[0,2/s\] in g(sλ). This raises two scenarios:
> > > 1. If s\<1: The full spectrum \[0,2\] is covered, and g(sλ) remains valid as a polynomial. (Please refer to Fig. 1 (2)).
> > > 2) If s\>1: Only the range \[0,2/s\] is valid; we suppress \[2/s, 2\] using a window function w(λ), where
> > >    $g_{truncated}​(sλ)=g(sλ)⋅w(λ)$, w(λ)= 1 at \[0, 2/s\], and w(λ)= 0 at (2/s, 2\],
> > >
> > >    Both g(sλ) and w(λ) are Chebyshev-approximable, keeping the full operation within O(N). (Please refer to Fig. 1(3) and Fig. 2)
> > >
> > > This method also removes the need for eigenvalue encoding and allows relaxation of the tight frame constraint (i.e., normalization), while preserving the core structure of WaveGC.
> > >
> > > **3\) Empirical Evaluation of Efficiency**
> > > We tested this simplified version on three short-range datasets against GPRGNN, BernNet, and UniFilter:
> > >
> > > * Runtime: As shown in *Table 1*, the simplified WaveGC achieves comparable training time to Fourier-based methods.
> > >
> > > * Accuracy: As shown in *Table 2*, it incurs only a small drop in performance, confirming that polynomial approximation remains effective even without eigendecomposition.
> > >
> > > This efficient variant of WaveGC opens new directions for practical deployment and future research. We will incorporate this approximation strategy into the manuscript and further refine it.

---

### Official Review · Reviewer_YKSE · 2025-03-12

**Overall Recommendation:** 3

**Summary:**

In this work, the authors introduce WaveGC, an innovative wavelet-based graph convolution approach featuring multi-resolution bases and a dense matrix kernel. They construct the necessary wavelet bases by leveraging Chebyshev polynomials of the first kind. For kernel implementation, they draw inspiration from AFNO, employing MLPs to create an effective kernel structure. The operation of WaveGC closely resembles FFT-based convolution. Theoretical analysis demonstrates that WaveGC effectively captures both short-range and long-range feature information. Comprehensive experimental results convincingly showcase the superior performance and effectiveness of WaveGC across various tasks.

**Claims And Evidence:**

Yes.

**Essential References Not Discussed:**

None.

**Experimental Designs Or Analyses:**

N/A

**Methods And Evaluation Criteria:**

Yes.

**Other Comments Or Suggestions:**

Although [1] was accepted by ICLR 2019, GWNN is not actually a GNN based on graph wavelet transform. First, graph wavelet transform requires the filter to be a band-pass filter, while GWNN uses the Heat Kernel, which is a low-pass filter. Second, graph wavelet transform is a discrete orthogonal transform, while $e^{-tL}$ is not a orthogonal matrix.

[1] Bingbing Xu, Huawei Shen, Qi Cao, Yunqi Qiu, & Xueqi Cheng (2019). Graph Wavelet Neural Network. In International Conference on Learning Representations.

**Other Strengths And Weaknesses:**

Strengths:
1. The authors prove that WaveGC can effectively captures both short-range and long-range feature information.

Weaknesses:
1. Eq. 1 is inappropriate. This is because in this formula, $\lambda$ represents the eigenvalue of the graph Laplacian. Eigenvalues are clearly discrete quantities and cannot be integrated.
2. Whether it's the graph Fourier transform or the graph wavelet transform, both are 1D discrete orthogonal transforms. For a 1D discrete orthogonal transform, a diagonal matrix rather than a full-size dense square matrix is sufficient as the kernel.

**Questions For Authors:**

1. In Eq. 9, what are the specific representation of $\mathbb{S}$ and $\mathbb{W}$? The explanation of different wavelet kernels is vague.

**Relation To Broader Scientific Literature:**

N/A

**Theoretical Claims:**

N/A

---

> ### Author Rebuttal · Authors · 2025-03-31
>
> We thank the reviewer for reviewing our work. Please, see below our answer to the raised comments/questions.
>
> > Q1: Eq. (1) is inappropriate.
>
> Eq. (1) describes requirements for general wavelets as introduced in Mallat’s book in 1999, not only graph wavelets, where g(λ = 0) = 0 is important for any data format. Thanks for your feedback, and we will change Eq. (1) to discrete version.
>
> > Q2: Why using a full matrix-valued kernel when a diagonal kernel is sufficient for 1D orthogonal transforms like the graph Fourier or wavelet transform?
>
> It is true that, in the context of classical 1D discrete orthogonal transforms (such as the Fourier or wavelet transform), convolution in the spectral domain can be represented using a **diagonal matrix** where each diagonal entry scales the corresponding frequency component. This is the foundation for many spectral GNNs that use vector-valued kernels (e.g., diag(θλ)).
>
> However, our design departs from this classical formulation by introducing a **matrix-valued kernel**, which enables richer transformations. Specifically:
> 1. **Diagonal kernels perform independent scaling across feature channels** for each frequency mode, limiting the ability to model interactions between features.
> 2. **Matrix-valued kernels**, in contrast, allow interaction within the feature dimension at each frequency mode. This is particularly useful in deep learning settings where features are multi-dimensional, and learning correlations between them is beneficial.
> 3. Furthermore, in our implementation, the same transformation (via a shared MLP) is applied across all spectral modes. This enforces a consistent filtering mechanism while keeping the parameter count manageable.
>
> Thus, while a diagonal kernel is mathematically sufficient for linear spectral filtering, we adopt a matrix-valued kernel to **increase expressiveness** and enable **learnable feature-channel interactions**, which empirically improves performance (as shown in Section 6.2). This design is inspired by similar motivations behind matrix-valued kernels in Fourier Neural Operators and Transformer models.
> We will revise the manuscript to more clearly explain this motivation and distinguish our approach from classical spectral filtering.
>
> > Q3: What are S and W in Eq. 9?
>
> S and W are the same as M in line 191 in the left column, representing different matrix-valued kernels. We will clarify this point.

---

### Official Review · Reviewer_JRdk · 2025-03-13

**Overall Recommendation:** 2

**Summary:**

This paper introduces WaveGC, a wavelet-based graph convolution network that integrates multi-resolution spectral bases with a matrix-valued filter kernel. It proposes graph wavelets by decomposing Chebyshev polynomials into odd and even terms and combining them with learnable coefficients, ensuring strict admissibility criteria and enhanced flexibility. The authors demonstrate that WaveGC effectively captures both extremely small and large scales, extending beyond previous graph wavelet theory. Experimental results show state-of-the-art performance in both short-range and long-range tasks.

## Update after rebuttal
I think this is a nice work but not quite up to my expectation. I will keep my original rating for now.

**Claims And Evidence:**

The paper provides theoretical and empirical support for WaveGC’s ability to capture both short-range and long-range information, as well as its superior performance over existing models. However, several claims require further validation.

1. The assertion that WaveGC covers extremely large scales ($s \rightarrow \infty$) is supported by theoretical proof but lacks empirical verification as there is no analysis of the actual learned scale values or their impact.

2. Since only a single scaling term is used for low-frequency information while multiple wavelet bases are employed, the authors justify the addition of an MPNN in parallel to augment low-frequency modeling. However, the scaling function already takes a value of 1 at $\lambda = 0$, and other wavelet bases can also learn low-frequency components near the origin. Fig.3 shows that $g(s_3 \lambda)$ retains a significant amount of low-frequency information, raising the question of whether the additional MPNN is truly necessary for enhancing low-frequency modeling. I think if WaveGC can achieve strong performance without MPNN, it would better demonstrate its adaptability.

**Essential References Not Discussed:**

The paper is referencing proper and essential papers.

**Experimental Designs Or Analyses:**

1. The ablation studies are presented on different datasets across Tab.4, Tab.5, and Tab.6, rather than consistently using the same set of datasets. This raises concerns whether each component consistently contributes to performance improvement. Additionally, in Appendix D.7, tight frames were not applied to ogbn-arxiv and Peptides-struct, and parameter sharing was not used for most long-range datasets. The paper claims these techniques as contributions for efficient computation and performance improvement. However, since strong performance is achieved without them on certain datasets, I am not so sure how to interpret this.
2. The paper states that, due to memory constraints, only the first 30% of eigenvalues and their corresponding eigenvectors were retained for short-range datasets. This approach likely results in a significant loss of high-frequency information, which is particularly important for short-range datasets. Given this, I have concerns on whether this strategy is appropriate unless kernels such as diffusion kernels are being tested.

**Methods And Evaluation Criteria:**

The proposed methods and evaluation criteria are generally well-aligned with the problem of spectral graph learning, as WaveGC is tested on both short-range and long-range benchmark datasets, demonstrating its adaptability. The use of various baselines is also appropriate. However, the rationale behind certain methodological choices is not so clear to me as,
1. The argument “powerful kernel with more parameters provide enough flexibility to adjust itself”  seems contradicting the later claim that parameter sharing is necessary due to excessive parameters.
2. The paper claims that the matrix-valued kernel allows various frequency modes to interact. However, since $F(X)$ has a dimension of $N \times d$ and the kernel M is $d \times d$, the operation MLP($F(X)$) appears to facilitate interaction between node features rather than frequency modes. Is this understanding correct?

**Other Comments Or Suggestions:**

Some minor comments would be:
1. In Eq. (3), shouldn't the dimension be Nxd instead of NxN?
2. The paper should states that the eigenvalues are restricted within [0,2] for normalized Laplacian to define $g(\lambda)$ is a strict band-pass filter in the range [0,2].
3. There is no explanation of what $H$ represents in Eq. (8).
4. There is a large blank space above "Other experiments" on page 8 and conclusion should be expanded.
5. In Table 3, the WaveGC result for Pf is reported as 69.10, but in Table 6, it appears as 69.01.

**Other Strengths And Weaknesses:**

Strength:
- The paper introduces a novel wavelet construction by decomposing Chebyshev polynomials into odd and even terms, ensuring wavelet admissibility.
- These wavelets are learnable and, together with the matrix-valued kernel, provide greater flexibility. It also includes a theoretical proof for long-range information capture and demonstrates strong performance through extensive experiments across various datasets and models, achieving state-of-the-art results.

Weakness:
- However, including the concerns raised above, I see the overall lack of clarity in the writing and the complex notations as weaknesses.

**Questions For Authors:**

- The notation in Eq. (6) seems unclear. Does $d$ here refer to the node feature dimension? According to Appendix C, the dimension of $\hat{Z}$ is given as $N \times (d+1)$, which does not align with the matrix operations.

**Relation To Broader Scientific Literature:**

This paper advances spectral graph convolution and graph wavelet transforms in SGWT (Hammond et al., 2011). It introduces a Chebyshev polynomial decomposition into odd and even terms, ensuring strict wavelet admissibility while enabling learnable wavelet construction. Additionally, it incorporates matrix-valued kernels, inspired by Fourier Neural Operators (Li et al., 2021), to enhance spectral filtering flexibility. The paper also expands on previous graph wavelet theory by providing a theoretical proof that WaveGC effectively captures both short-and-long range information from the perspective of information mixing (Di Giovanni et al., 2023), addressing a limitation in prior work (Hammond et al., 2011) that focused only on small-scale localization.

**Theoretical Claims:**

I have not verified the validity of Theorem 4.2 as it is in the appendix.

---

> ### Author Rebuttal · Authors · 2025-03-31
>
> We appreciate the review provided. Please, find below our clarifications on the points raised, and the link to the figures & tables: https://drive.google.com/file/d/1DALF7e1t6O4SSMUkvfi2euMufIpNJARv.
>
> > Q1: Analysis of learned scales and experimental impacts.
>
> 1) Learned Scales and Receptive Fields
>
> Figure 1 visualizes the largest learned scales and corresponding receptive fields for Ps and VOC, using the heuristic: Given a wavelet Ψ(sλ), node j lies in the receptive field of node i if |Ψ(sλ)[i, j]| > 0.1 × max(|Ψ(sλ)|).
>
> Under this criterion, Ps exhibits a significantly larger receptive field (scale = 9.48), aligning with its inherently longer average shortest-path distances. Table 1 further confirms this through average and max receptive field sizes across all nodes.
>
> 2) Behavior at Extremely Large Scales (s → ∞)
>
> We increased the predefined scale vector $\bar{s}$ in Eq. (7) for Pf to be (10, 100, 1000). This vector determines the upper bound of the learnable scale range. As shown in Figure 2, the receptive field expands from local (s = 9.17) to global (s = 988.24), empirically supporting our theoretical claim that WaveGC captures long-range dependencies as 𝑠→∞.
>
> > Q2: Verify the necessity of MPNN for low-frequency information.
>
> As shown in Table 2 (w/o MPNN), WaveGC remains competitive without the MPNN branch, though with some performance drop—demonstrating its adaptability while also showing the benefit of MPNN.
>
> Wavelet bases are inherently localized and suppress low-frequency signals, enabling node-specific filtering for local structure modeling. However, global community-level information—captured by low-frequency components—is still important. A single scaling function h(λ) appears insufficient to model this alone, as reflected by the performance drop without MPNN. Therefore, we argue that including an MPNN branch complements the wavelet by restoring low-frequency (community-level) signals.
>
> Notably, removing the wavelet branch (w/o wavelet) leads to a larger performance drop, confirming that wavelets are the primary contributor, with MPNN as a helpful supplement.
>
> > Q3: Description on kernel parameters.
>
> Our goal is to balance expressiveness and efficiency. While matrix-valued kernels offer greater capacity than vector-valued ones, assigning separate matrices per frequency (as in FNO) leads to excessive parameters and risk of overfitting. To mitigate this, we use a shared MLP across frequencies, which retains modeling power while keeping parameter count low. As shown in Tables 3 and 4, this strategy improves performance with manageable complexity. We will revise the text to clarify this trade-off.
>
> > Q4: Usage of tight frames and parameter sharing.
>
> 1) **Parameter sharing**. In Appendix D.7, “parameter sharing” refers to sharing parameters across stacked WaveGC+MPNN blocks (Fig. 2 in the paper). Most datasets use independent parameters per layer, which performed better empirically. This is unrelated to the matrix kernel’s weight-sharing, which is always applied. We will revise the wording to clarify this.
>
> 2) **Tight frames**. The tight frame condition simplifies Eq. (9), but requires normalizing the scaling and wavelet bases, which may restrict model expressiveness. For ogbn-arxiv and Peptides-struct, we relaxed this constraint (i.e., omitted normalization) to prioritize performance. Importantly, **all datasets** still used Eq. (9), so the computational efficiency remained unaffected.
>
> > Q5: Does MLP(F(X)) facilitate interaction between node features?
>
> It is correct that the matrix-valued kernel (via MLP) operates within the feature dimension of each spectral component, not directly across frequency modes. Each row in $F(X)\in \mathbb{R}^{N(J+1) \times d}$ is transformed independently by the shared MLP.
>
> However, since all spectral modes share the same MLP, the model learns a **unified transformation** that generalizes across modes. This shared design **indirectly couples the modes**, as the MLP must accommodate diverse spectral inputs. We will revise the text to clarify this mechanism.
>
> > Q6: Rationality of only keeping the first 30% graph spectrum.
>
> Following your suggestion, we tested three types of spectral filters: 1) **Low-frequency (diffusion) kernel**: g1(λ)=exp⁡(−β⋅λ), 2) **High-frequency kernel**: g2(λ)=exp⁡(−β⋅(2−λ)) and 3) **Combined kernel**: g3(λ)=g1(λ)+g2(λ). Applied using R=Ug(λ)U⊤, the results (Table 6) show that g1 consistently outperformed the others across four short-range datasets. In contrast, g2 performed poorly, and g3 showed mixed results. On ogbn-arxiv, the diffusion kernel caused OOM.
>
> These findings support our choice to retain only the first 30% of the spectrum: it captures the most relevant low-frequency signals while reducing computational cost and noise from high-frequency parts.
>
> > Q7: Use the same datasets across experiments.
>
> We have supplemented experiments as in Tab. 3, 4 and 5 in the link. New results did not overturn the conclusions in the paper.

---

> > ### Comment · Reviewer_JRdk · 2025-04-04
> >
> > I think the authors may have misunderstood my question regarding the use of only the first 30% of the eigenvalues and I did not ask for additional experiment. In fact, this new result raises me doubt as the 3) combined case is not best, i.e., combining representations from low-frequency and high-frequency does not improve the result from using the 1) diffusion kernel only. As wavelets are natural band-pass filters, this only means that wavelets as band-pass filtering harms the performance and only a scaling function should be adopted.
> >
> > What I wondered in the first place was that, focusing on the small eigenvalues limits the behavior of wavelets as band-pass filters as the $\lambda$s used in the experiment will be distributed mostly near the origin rather than 2, and thus $g(\lambda) \approx 0$.

---

> > > ### Author Response · Authors · 2025-04-06
> > >
> > > Thank you for taking the time to read our response. We apologize for our misunderstanding of the question. Below, we clarify why focusing on small eigenvalues—i.e., retaining only the first 30% of the spectrum—does not limit the behavior of wavelets. We present our explanation from both **spectral** and **spatial** perspectives:
> > > 1) **Spectral perspective: Truncated eigenvalues still allow wavelets to capture meaningful low-frequency signals.**
> > >
> > > As shown in Fig. 5 (Appendix D.3), the wavelet function g(sλ) retains non-trivial amplitudes within the first 30% domain. While g(λ)≈0, the retained spectral range is sufficiently broad to allow the wavelets to operate effectively.
> > > This aligns with our experiments showing that low-frequency components are most beneficial for short-range datasets, while high frequencies may harm performance. Therefore, even in the truncated setting, both the scaling function h(λ) and wavelet function g(λ) contribute meaningfully to low-frequency modeling.
> > >
> > > 2) **Spatial perspective: Spectral truncation mimics larger scales, expanding receptive fields and enabling higher-order aggregation.**
> > >
> > > In Fig. 5(b) (Appendix D.3), we observe that truncating the spectrum mimics the effect of using a larger wavelet scale s>1 (see also Fig. 1(3) in https://drive.google.com/file/d/1cLakrmKX0nOlcyv_RC2_e38npfxiyKlJ/view?usp=sharing), which reduces the effective spectral range and increases the spatial receptive field.
> > >
> > > This effect is visually confirmed in Fig. 4(c) and 5(c) (Appendix D.3), where the receptive fields become noticeably larger after truncation. Thus, even on short-range datasets, the wavelet branch captures valuable higher-order information that complements local aggregation from MPNN. This complementary role is further validated by the performance drop observed in the ablation study (Table 2 in https://drive.google.com/file/d/1DALF7e1t6O4SSMUkvfi2euMufIpNJARv.) when wavelets are removed.
> > >
> > > Overall, spectral truncation does not impair wavelet behavior; instead, it supports effective low-frequency modeling while also enhancing spatial coverage. We will revise the manuscript to better explain this nuanced interaction.

---

### Decision · Program_Chairs · 2025-05-01

**Decision:**

Accept (poster)

**Comment:**

This paper proposes a wavelet-based graph convolution architecture (WaveGC) designed to integrate learnable spectral wavelet bases with matrix-valued kernels, resulting in more expressive and localized graph representation learning. The method introduces a novel decomposition of Chebyshev polynomials into odd and even components, ensuring admissibility and multi-scale expressiveness. The model is shown to outperform existing spectral methods, particularly in long-range and heterophilous settings. The work is supported by both theoretical insights and empirical validations across a broad range of benchmarks.

There is a range of reviewer sentiment across the four reviews, though the discussion and rebuttal process was constructive. The authors responded to all concerns in satisfactory ways, and demonstrated a willingness to clarify design decisions and revise the manuscript accordingly.

**Weak Accepts:**

Reviewers 5Zza, and YKSE find the contribution meaningful and well-supported. Reviewer 5Zza highlights that many concerns raised in a prior review round have been addressed and praises the theoretical rigor and empirical validation. Reviewer YKSE raised some technical questions on formulation and implementation details (e.g., kernel choice and matrix structure), which were answered to their satisfaction in the rebuttal.

**Weak Rejects:**

Reviewers 4MPc and JRdk express concerns around either comparative analysis or interpretability of results. In particular, 4MPc critiques the absence of Fourier-basis baselines and the scalability of WaveGC. In response, the authors ran new experiments comparing against GPR-GNN, BernNet, UniFilter, and S²GCN, showing that WaveGC consistently outperforms them, particularly in long-range settings. The authors also proposed a polynomial-based approximation to reduce complexity to O(N), directly addressing scaling concerns. Reviewer JRdk initially questioned the necessity of MPNN for low-frequency recovery and the truncation of eigenvalues; these were answered through both visualization and analytical explanations. While JRdk remained skeptical about the spectral range truncation, the authors presented both spectral and spatial perspectives to support their design.

Reviewer 4MPc remained concerned about scalability and still found the improvements over Fourier-based methods to be insufficiently compelling. Yet, to my understanding, the new experiments provided by the authors in the rebuttal do address these concerns.

**Summary**

The authors’ rebuttal was comprehensive and collegial throughout: In addition to running new comparisons, they clarified theoretical justifications (e.g., admissibility, receptive field behavior, and MPNN integration) and responded to formatting and clarity suggestions. WaveGC is an interesting contribution to spectral graph learning with a new, well-motivated architecture and empirical strength across diverse tasks. The addition of a computationally efficient variant further improves the paper’s relevance. Assuming the authors follow through with their promised revisions, this work would be a good addition to ICML.